# Anterior and posterior retrosplenial cortex form distinct visuospatial circuits in the mouse

Yu-Ting Wei [1,2] ✉, João Couto[1,3], Fabian Kloosterman [1,4,6] & Vincent Bonin [1,2,5,6] ✉

The retrosplenial cortex (RSC) integrates sensory and mnemonic information to support spatial orientation and navigation, yet how visuospatial processing differs across its subregions remains unclear. Here, we combined cellular imaging in navigating mice with brain-wide anatomical input tracing to characterize how multimodal sensory and positional signals are integrated along the anterior–posterior axis of dorsal RSC. We identified consistent differences between anterior and posterior subregions in both functional response properties and long-range connectivity. Anterior RSC neurons displayed sharper and more reliable position tuning during tactile-cued navigation and preferential sensitivity to fast, low-spatial-frequency visual motion. In contrast, posterior RSC neurons showed broader position selectivity, stronger responses to slow, high-spatial-frequency visual patterns, and enhanced tuning in visually immersive virtual environments. Consistent with these differences, anterior RSC received denser projections from motor, somatosensory, and parietal areas, whereas posterior RSC received stronger input from primary and posteromedial visual cortices. Together, these findings identify an anterior–posterior functional gradient in RSC, with subregions differing in how they integrate sensory and positional signals during navigation.

The retrosplenial cortex (RSC) is an important hub for integrating sensory, mnemonic, and contextual information to support navigation, learning, and memory[1–5]. Through its extensive connections with sensory, motor, thalamic, and hippocampal circuits[6–9], the rodent RSC is well positioned to transform multimodal information into behaviorally relevant representations that guide navigation and behavior[10–13]. Despite this central role, the organizational principles governing these multimodal information flows within the RSC remain poorly understood.

During spatial navigation, the brain must continuously update the animal's position and orientation by integrating self-motion cues with external sensory landmarks[14]. Rodent RSC neurons encode diverse spatial variables, including landmarks[11,15], reward locations[16],

head direction[17–19], and egocentric boundaries[20–22], highlighting its potential role in aligning internal representations with external space. Such representations are important for spatial learning and memory, and RSC lesions or inactivation produce profound navigational impairment[15,23,24]. Vision plays an important stabilizing role: through reciprocal connections with visual and hippocampal systems, the RSC may link sensory cues with long-term spatial memory. RSC neurons exhibit both visually evoked activity and position-selective responses[25–28], suggesting that this region serves as an interface between sensory perception and internal spatial maps. Together, these observations highlight the diversity of spatial signals encoded in RSC but do not explain how these signals are integrated within the RSC.

[1]VIB-KU Leuven Center for Neuroscience/NERF, Leuven, Belgium. [2]Department of Biology & Leuven Brain Institute, KU Leuven, Leuven, Belgium. [3]Department of Neurobiology, University of California, Los Angeles, CA, USA. [4]Faculty of Psychology & Educational Sciences, KU Leuven, Leuven, Belgium. [5]Department of Biosystems, KU Leuven, Leuven, Belgium. [6]These authors jointly supervised this work. ✉e-mail: ytsimon2004@gmail.com; vincent.bonin@kuleuven.be

Anatomically, RSC spans the anterior–posterior extent of the dorsal posterior cortex and contains dorsal (RSCd) and ventral (RSCv) subdivisions, which differ in lamination architecture and connectivity[9,29,30] and support distinct memory-related behaviors[31,32]. While these dorsal–ventral subdivisions have been well characterized, functional and anatomical distinctions along the anterior–posterior axis are less well understood. Posterior RSC has been implicated in contextual and fear-related memory encoding[33] and receives denser inputs from visual cortices, supporting its role in scene and context processing[28,34]. In contrast, anterior RSC shows stronger coupling with motor areas, consistent with its involvement in action planning and motor integration[7,34]. How these anterior–posterior differences shape visuospatial processing during navigation—and how they relate to long-range connectivity—remains unclear.

Here, we combined two-photon calcium imaging during navigation with brain-wide retrograde tracing to characterize how visual and position-related signals are integrated along the anterior–posterior axis of the mouse RSC. We show that anterior and posterior RSC subregions exhibit distinct specializations: anterior RSC neurons show more robust and reliable position tuning and are preferentially responsive to fast, low–spatial-frequency visual motion, whereas posterior RSC neurons are tuned to slow, high–spatial-frequency stimuli and display stronger position coding when visual landmarks are prominent. These functional distinctions were mirrored by differences in long-range wiring, with sensorimotor and parietal projections enriched in anterior RSC and visual cortical projections enriched in posterior RSC. Together, our results reveal an organization in which distinct RSC subregions integrate distinct combinations of sensory cues to support navigation.

## Results

We combined functional cellular imaging with long-range anatomical input tracing to study differences in neuronal response properties across subregions of the mouse retrosplenial cortex (RSC) and its brain-wide inputs (Fig. 1A). We focused on anterior and posterior RSC subregions, which differ in lamination architecture and connectivity[9,29,30], support distinct behaviors[31,32], and thus may show

distinct neuronal responses and input patterns. We first characterized responses across anterior–posterior RSC subregions and then investigated brain-wide inputs to these subregions. To characterize functional properties, we initially investigated how visual and position-related signals are encoded across neuronal populations in RSCd subregions (Fig. 1B).

GCaMP6s-expressing mice were implanted with a chronic cranial window and trained to run on a linear treadmill. While the animals ran, a water reward was delivered at a fixed location, and tactile landmarks were affixed to the treadmill belt, providing salient cues about position along the track. This cued-belt paradigm reliably elicits position-related activity in hippocampal and retrosplenial areas[27,35,36]. The cellular imaging targeted layer 2/3 neurons in dorsal RSC (RSCd). Sampling of deeper layers and ventral RSC was limited. Using two mouse lines, we recorded from large populations of cortical excitatory neurons (Thy1-GCaMP6s, $n = 7$ animals; and CaMKII-tTA × TRE-GCaMP6s, $n = 8$ animals; Supplementary Tables 1 and 2, see "Methods"). Each recording began with a 15-min period during which the visual display showed a uniform gray screen. This was followed by a 30-min block of full-field visual stimulation.

The primary data set characterized RSC activity while animals ran on a treadmill with tactile landmarks. In subsequent experiments, the cues were removed, and the setup was placed either in darkness or in a visually immersive virtual reality (VR) corridor. Throughout these experiments, fields of view were placed over anterior and posterior RSC subregions at stereotaxic coordinates ranging from −1.04 to −2.38 mm (anterior) and −2.32 to −4.17 mm (posterior) relative to Bregma (Fig. 1B, C). Together, this enabled controlled characterization of visual and navigation-related signals across anterior and posterior subregions of RSCd.

### Position coding varies along the anterior–posterior RSC axis

We first quantified single-neuron position selectivity across anterior and posterior RSC subregions using occupancy-normalized, deconvolved calcium activity and a shuffle-based statistical test (Fig. 2A, left and Supplementary Fig. 1). For each neuron, we quantified the fraction of position-selective neurons, tuning reliability, peak response

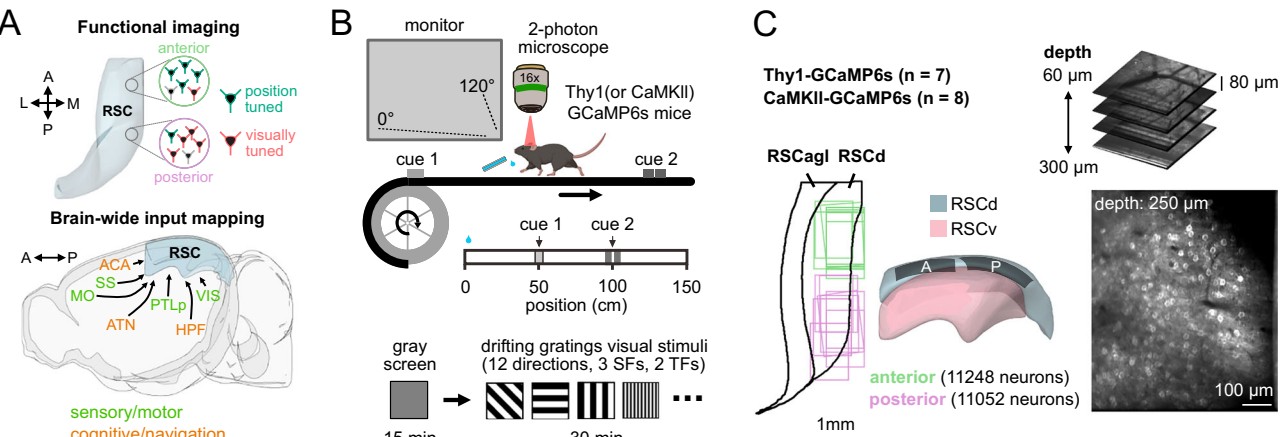

**Fig. 1 | Combined functional cellular imaging and brain-wide input mapping for investigating regional specialization in mouse RSC. A** Schematic of the experimental framework for investigating functional and anatomical specialization along the anterior-posterior axis of the RSC. Anatomical directions: anterior (A), posterior (P), lateral (L), and medial (M). Image created using the brainrender open-source Python library. **B** Top: Head-fixed two-photon imaging setup. Mice ran on a linear treadmill with two position cues (cues 1 and 2, at 50 cm and 100 cm) and received a water reward at a fixed location (0 cm) on each lap. A monitor was placed 18 cm from the right eye, covering 120° × 80° of the right visual field. Bottom: Each session began with a 15-min gray screen to measure position-related activity, followed

by a 30-min visual stimulation block with randomized full-field drifting gratings (12 directions, 3 spatial frequencies, 2 temporal frequencies). Mouse cartoon created in BioRender. Wei, Y. (2026) https://BioRender.com/c0ric0k. **C** Locations of imaged fields of view (FOVs) and estimated imaging depth across the anterior–posterior extent of RSC. Imaging primarily sampled layer 2/3 neurons in dorsal RSC (RSCd), with minor coverage of deeper layers and ventral RSC (RSCv). Each FOV measured 892 × 667 μm. In a subset of experiments, volumetric scanning was performed using an electrically tunable lens. $n = 10$ fields of view per group (anterior or posterior RSC) from 8 animals. Image created using the brainrender open-source Python library.

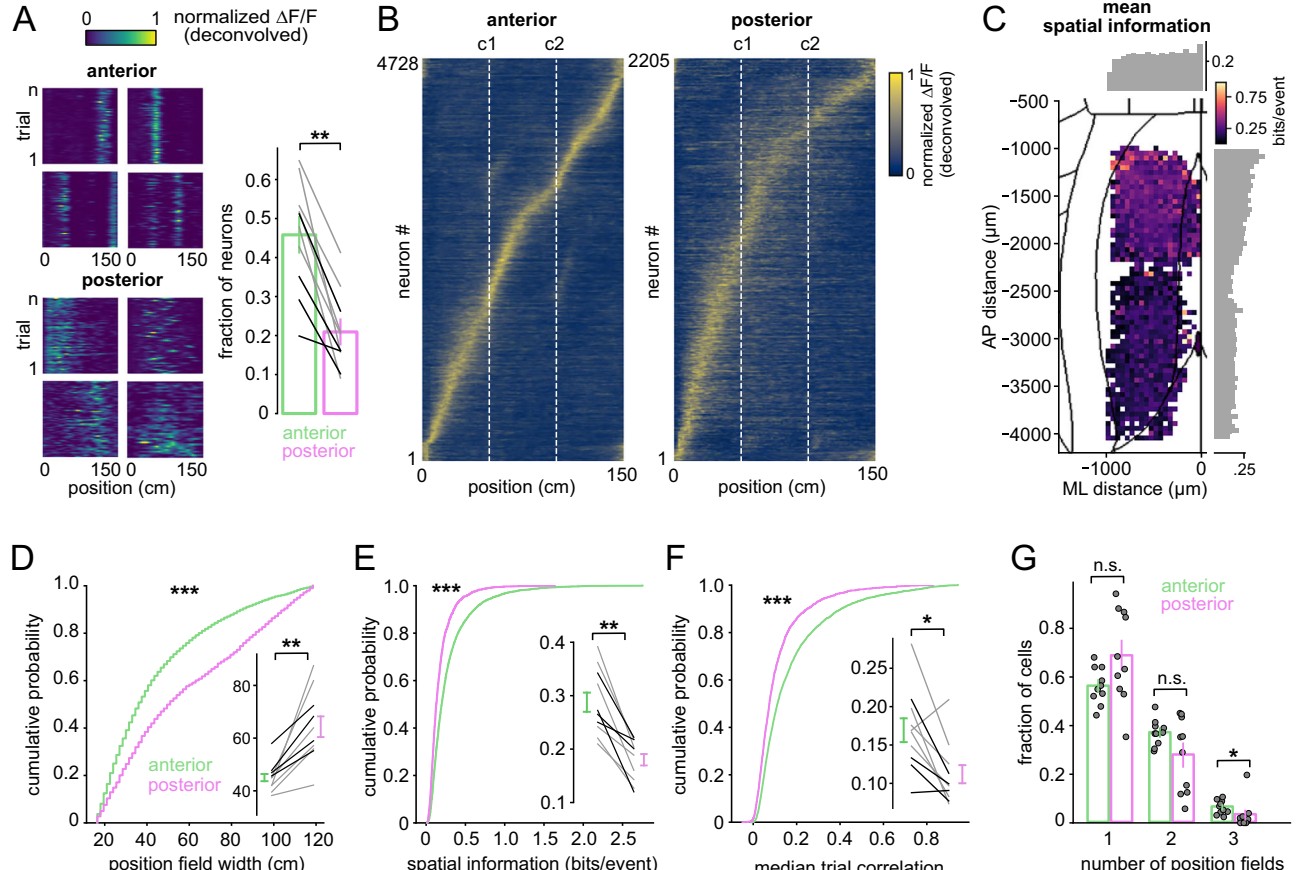

**Fig. 2 | Position tuning in anterior and posterior RSC differs in reliability and spatial scales. A** Left: Examples of normalized deconvolved calcium activity ($\Delta F/F_0$) from individual position-tuned RSC neurons, plotted as a function of the animal's position across trials. Right: Proportion of position-tuned neurons in anterior versus posterior RSC. Gray/black connected lines indicate individual imaging sessions for the same animal (gray: Thy1-GCaMP6s; black: CaMKII-GCaMP6). Vertical bars indicate mean ± SEM. Two-sided Mann–Whitney U test, $p = 0.0017$; $n = 10$ sessions per group from 8 animals. **B** Cross-validated, trial-averaged deconvolved $\Delta F/F_0$ activity for all position-tuned neurons in anterior and posterior RSC. Neurons are sorted by the position of peak activity computed from half of the data and plotted with the remaining half ($n = 10$ sessions). **C** Distribution of spatial information across position-tuned neurons, mapped onto the dorsal cortical surface. The heatmap shows mean spatial information at each cortical location. Histograms display averages across bins along the anterior–posterior and medial–lateral axes ($n = 10$ sessions). Cumulative distributions comparing **D** tuning field width, **E** spatial information (bits/event), and **F** trial-to-trial correlation coefficients between anterior and posterior RSC neurons. Animal- and recording-wise mean ± SEM values are shown with connected lines (gray: Thy1-GCaMP6s; black: CaMKII-GCaMP6). Statistical comparisons were performed using two-sided Kolmogorov–Smirnov (KS) tests on pooled data (4728 vs. 2205 neurons) and Two-sided Mann–Whitney U tests on per-session averages ($n = 10$ sessions): KS test vs. Mann–Whitney U test; tuning field width, $p = 1.8e-70$ vs. $p = 0.002$; spatial information, $p = 4.2e-62$; $p < 0.001$; median trial correlation, $p = 7.6e-46$ vs. $p = 0.017$. **G** Fraction of neurons with one, two, or three position fields. Vertical bars indicate mean ± SEM across sessions. Two-sided Mann–Whitney tests: 1 field, $p = 0.10$; 2 fields, $p = 0.21$; 3 fields, $p = 0.006$ ($n = 10$ sessions).

amplitude, tuning-field width, the number of fields, and spatial information (see "Methods"). We compared anterior and posterior FOVs recorded during treadmill locomotion with a uniform gray screen.

Position-tuned responses were observed in both anterior and posterior RSC, but the prevalence of position-tuned neurons was higher in anterior RSCd (Fig. 2A, right; anterior vs. posterior: 45 ± 5 % vs. 21 ± 3 %; mean ± SEM; $p = 0.0017$; two-sided Mann–Whitney U test; $n = 10$ sessions per group from 8 animals, for pooled data statistical tests, see figure captions). Neurons in anterior RSC exhibited sharper tuning (tuning field width: 45 ± 1.7 cm vs. 64.3 ± 4.2 cm; $p = 0.002$), carried more spatial information (0.28 ± 0.02 vs. 0.18 ± 0.01 bits/event; $p = 3.5 \times 10^{-9}$), and showed more reliable tuning (trial-to-trial correlation: 0.17 ± 0.02 vs. 0.11 ± 0.01; $p = 0.017$; Fig. 2B–F). Spatial information values computed from average tuning curves declined progressively along the anterior-posterior axis, particularly within anterior imaging RSC FOVs (Fig. 2C). The number of position fields per neuron was similar across subregions, though anterior RSC FOVs contained a slightly higher fraction of multi-field neurons (Fig. 2G).

Single-field responses dominated in both regions (56.4 ± 2.4% vs. 68.9 ± 6%; $p = 0.10$), while two-field responses were somewhat more common anteriorly (37.1 ± 1.7% vs. 27.9 ± 4.8%; $p = 0.21$). A small fraction of neurons displayed three fields, with a higher proportion in anterior RSC FOVs (6.4 ± 0.9% vs. 3.2 ± 1.9%; $p = 0.006$).

We next characterized population-level encoding of position in anterior and posterior RSC. Anterior RSC populations exhibited clearer sequential activity during treadmill running than posterior subregions (Fig. 3A, B, top). Bayesian decoding analysis revealed more robust representations in anterior RSC than in posterior RSC (mean decoding error: 5.2 ± 0.7 cm vs. 10.4 ± 1.2 cm; $p = 0.0017$; Fig. 3A, B, bottom; Fig. 3C, D). Differences in response amplitudes were also observed, with neurons in anterior RSC showing higher response amplitudes during treadmill running than those in posterior RSC (mean peak amplitudes: 110 ± 4.1% vs. 95.9 ± 3.9%; $p = 0.037$; Fig. 3E). Taken together, single-cell and population analyses reveal systematic differences in position-related representations between anterior and posterior RSCd during treadmill locomotion.

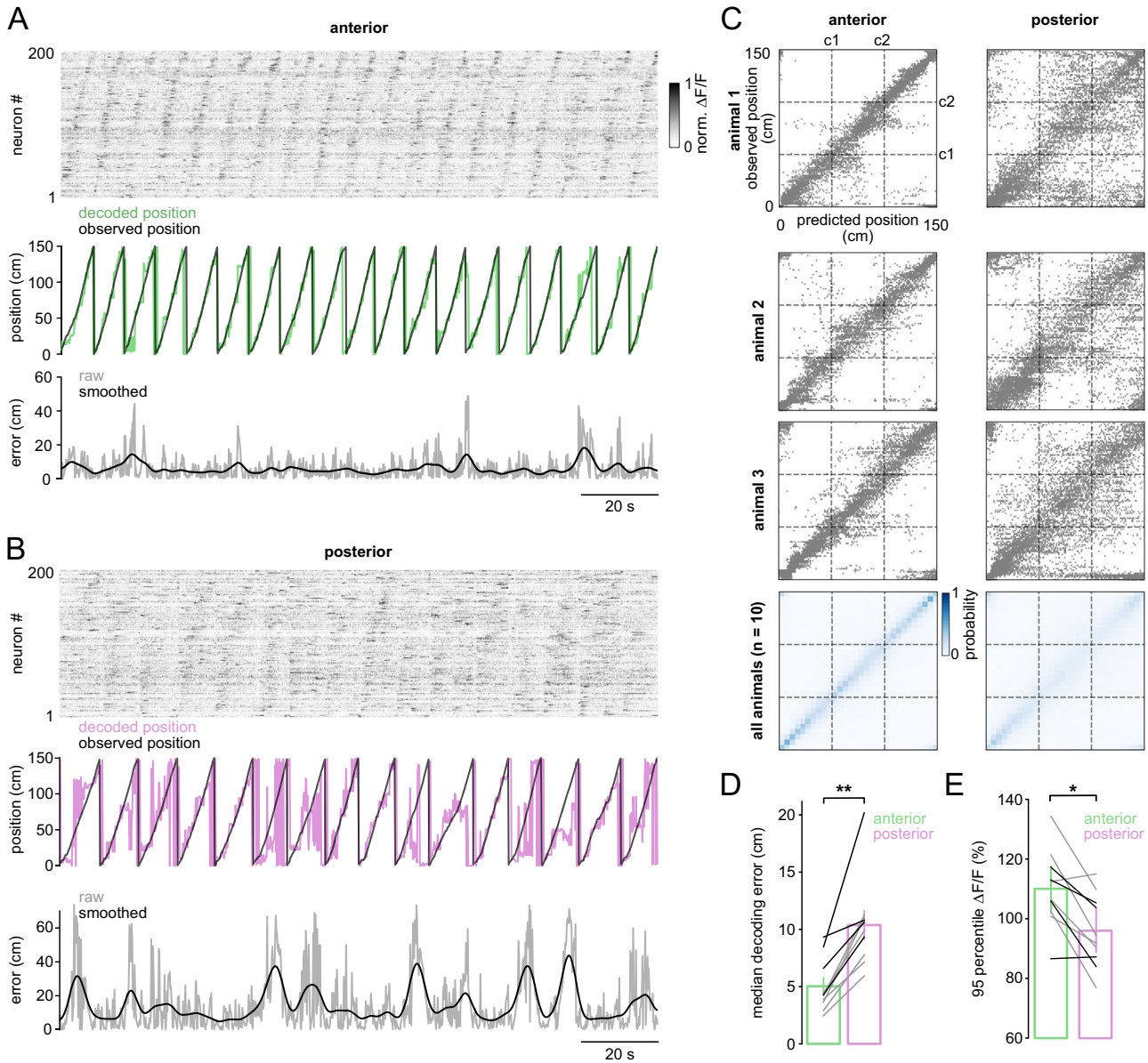

**Fig. 3 | Population decoding shows an anterior–posterior gradient in position activity in cued environments. A**, **B** Top: Normalized calcium activity ($\Delta F/F_0$) from 200 randomly selected RSC neurons, sorted by the position of peak calcium activity. Middle: Observed (black) and decoded (green and magenta) position over time for the same session. Bottom: Decoding error (absolute difference between observed and predicted position), smoothed with a Gaussian kernel ($\sigma = 10$ time bins). **C** Top: Confusion matrices showing predicted versus observed position from three animals. Bottom: Mean confusion matrix calculated across animals and recording sessions ($n = 10$ sessions per group from 8 animals). Cue locations (c1 and c2) are marked with dotted lines. **D** Median decoding error in anterior versus posterior RSC. Vertical bars indicate mean ± SEM across sessions; each connected line indicates an individual animal (gray: Thy1-GCaMP6s; black: CaMKII-GCaMP6). Two-sided Mann–Whitney U test, $p = 0.0017$ ($n = 10$ sessions). **E** 95th percentile $\Delta F/F_0$ amplitude in anterior versus posterior RSC. Vertical bars represent mean ± SEM; each connected line indicates an individual animal (gray: Thy1-GCaMP6s; black: CaMKII-GCaMP6). Two-sided Mann–Whitney U test, $p = 0.037$ ($n = 10$ sessions).

## Anterior and posterior RSC rely on different sensory cues for position coding

Position encoding depends on multiple modalities, including visual, somatosensory, and self-motion cues. To determine whether anterior and posterior RSC subregions rely on similar sensory inputs to generate position-related activity, we assessed position tuning under targeted manipulations of tactile and visual inputs.

We first tested the contribution of somatosensory inputs to position coding by removing the tactile landmarks from the treadmill belt (Supplementary Fig. 2A–C; $n = 8$ anterior vs. 6 posterior sessions from 9 animals). Eliminating these

somatosensory cues reduced the fraction of position-tuned neurons in anterior RSC (45 ± 5% to 35 ± 5%; mean ± SEM; $p = 0.031$), whereas the number of position-tuned cells in posterior RSC remained unchanged (21 ± 3% to 22 ± 7%; mean ± SEM; $p = 0.79$). This result suggests that anterior RSC relies more strongly on tactile inputs to maintain stable position representations.

We then tested next tested the role of visual input by recording neural activity while animals ran on the treadmill in darkness (Supplementary Fig. 3; $n = 9$ sessions from 7 animals). Bayesian decoding analysis revealed that turning off ambient light reduced the reliability of position responses in both RSC subregions (average decoding error

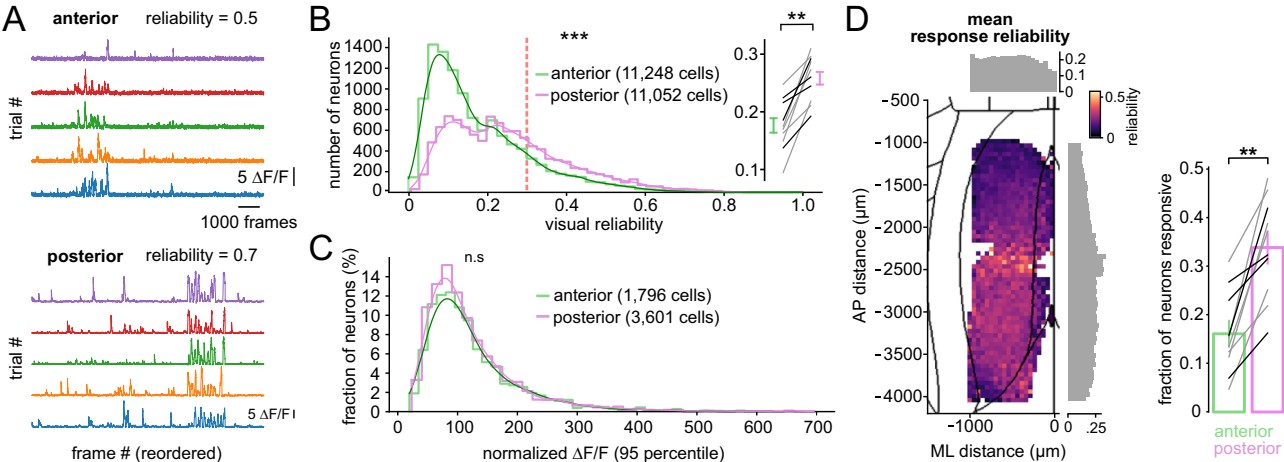

**Fig. 4 | Different degrees of visual responsiveness across anterior and posterior RSC subregions. A** Example $\Delta F/F_0$ time courses used to assess visual reliability. Responses to the stimuli were de-randomized and sorted by trial (rows). Reliability index ($r$) was computed as the 75th percentile of trial-to-trial Pearson correlation coefficients. **B** Distribution of visual reliability indices for all recorded neurons across anterior and posterior RSC. The dotted line indicates the threshold (0.3) used to classify a neuron as visually responsive. Kolmogorov-Smirnov test; $p = 5.7$e-264; 11,248 (anterior) vs. 11,052 (posterior) neurons. Connected lines show per-session averages from individual animals (gray: Thy1-GCaMP6s; black: CaMKII-GCaMP6s); bars indicate mean ± SEM. Two-sided Mann–Whitney U test; $p = 0.002$ ($n = 10$ sessions per group from 8 animals). **C** Distribution of maximal $\Delta F/F_0$

amplitudes (95th percentile across individual trials) for visually responsive neurons. Two-sided Kolmogorov-Smirnov test; $p = 0.08$; $n = 1796$ (anterior) vs. 3601 (posterior) neurons ($n = 10$ sessions). **D** Left: Spatial distribution of mean visual reliability across the dorsal cortical surface. The heatmap indicates the average visual reliability at each cortical location. Histograms show visual reliability averaged across spatial bins along the anterior–posterior and medial–lateral axes. Right: Proportion of visually responsive neurons in anterior versus posterior RSC. Vertical bars indicate mean ± SEM across sessions. Each connected line represents an individual animal. Two-sided Mann–Whitney U test; $p = 0.0017$ ($n = 10$ sessions, gray: Thy1-GCaMP6s; black: CaMKII-GCaMP6).

across subregions: 7.8 ± 0.9 cm to 15.7 ± 1.6 cm; $p = 0.0039$; $n = 9$ sessions, Supplementary Fig. 3A, B). However, the anterior–posterior gradient in responses persisted. Even in the absence of visual input, posterior RSC showed a trend toward larger decoding errors as compared with the anterior subregion (13.4 ± 0.7 cm vs. 20.1 ± 3.8 cm; $p = 0.16$, anterior vs. posterior, $n = 6$ vs. 3 sessions; Supplementary Fig. 3B). Posterior RSC also showed a lower fraction of position-tuned neurons as compared to anterior RSC (20 ± 2.5% vs. 7.6 ± 0.8%; $p = 0.024$; anterior vs. posterior, $n = 6$ vs. 3 sessions; Supplementary Fig. 3C). The reliability gradient of position-related activity was also partially maintained (Supplementary Fig. 3D–F). Together, these results indicate visual inputs influence position coding in both anterior and posterior RSC.

Finally, we asked whether a visually-immersive VR task would differentially affect position coding in RSC subregions. To address this, we recorded RSC activity while animals navigated a VR corridor with salient visual landmarks (Supplementary Fig. 4A, B, $n = 3$ animals). After learning the task, as indicated by anticipatory licking near the reward zone (Supplementary Fig. 4C), both anterior and posterior RSC subregions exhibited clear single-cell position tuning and robust population representations of position (Supplementary Fig. 4D, E). Notably, posterior RSC showed improved tuning relative to that observed during navigation of the tactile-cued treadmill (Fig. 2A). After excluding neurons with peak responses near the end of the corridor (>200 cm), the proportions of position-tuned neurons in anterior and posterior RSC were comparable (44 ± 12% vs. 40 ± 8%; $n = 3$ sessions per group from 3 animals; Supplementary Fig. 4F–H).

Together, these results demonstrate distinct sensory dependencies along the anterior–posterior axis of RSC: anterior RSC relies more on somatosensory tactile cues, whereas posterior RSC exhibits enhanced position tuning in the presence of structured visual input. This pattern is consistent with differential integration of contextual information by the two subregions during navigation. To further characterize this anterior–posterior specialization, we next examined visually evoked responses independent of position coding.

## Visual response properties differ between anterior and posterior RSC

The RSC receives input from both primary and higher-order visual cortex and displays visually evoked activity[25,37]. Furthermore, anterior and posterior subregions of the visual cortex show distinct visual tuning preferences[38–41]. To assess whether RSC subregions show visual specialization, we examined responses to visual stimulation. To characterize the large-scale organization of these responses, we performed widefield calcium imaging, which revealed a coarse retinotopic organization along the anterior–posterior axis: posterior RSC responded preferentially to stimuli in the nasal upper visual field, whereas anterior RSC showed weaker activation and was preferentially driven by stimuli in the nasal lower visual field (Supplementary Fig. 5 and Supplementary Movie 1).

We next characterized the visual responses of RSC neurons at cellular resolution. To this end, we presented full-field drifting gratings spanning six orientations, twelve directions, three spatial frequencies (0.04–0.16 cpd), and two temporal frequencies (1 and 4 Hz) while the animals ran on the treadmill. Visually responsive neurons were defined by a median trial-to-trial correlation greater than 0.3 (Fig. 4A and Supplementary Fig. 6). This criterion allowed us to focus subsequent analyses on neurons with consistent visually evoked responses. Consistent with the widefield measurements, we observed cellular differences in visual responsiveness across RSC subregions. Posterior RSC contained a larger proportion of visually responsive neurons and displayed higher visual reliability than anterior RSC (16.1 ± 2.7% vs. 33.8 ± 3.3%, $p = 0.0017$; reliability: 0.18 ± 0.04 vs. 0.26 ± 0.04, $p = 0.0022$; two-sided Mann–Whitney U test; $n = 10$ sessions per group from 8 animals; Fig. 4B–D).

We found differences in spatial and temporal frequency tuning across anterior and posterior RSC subregions. Anterior RSC neurons preferred low spatial frequency, high temporal frequency stimuli (e.g., 0.04 cpd at 4 Hz), whereas posterior RSC neurons were biased toward high spatial frequency, low temporal frequency stimuli (e.g., 0.16 cpd at 1 Hz; Fig. 5A–D). These differences were reflected in the distributions of neurons responsive to each SF/TF combination (e.g., 0.04 cpd at

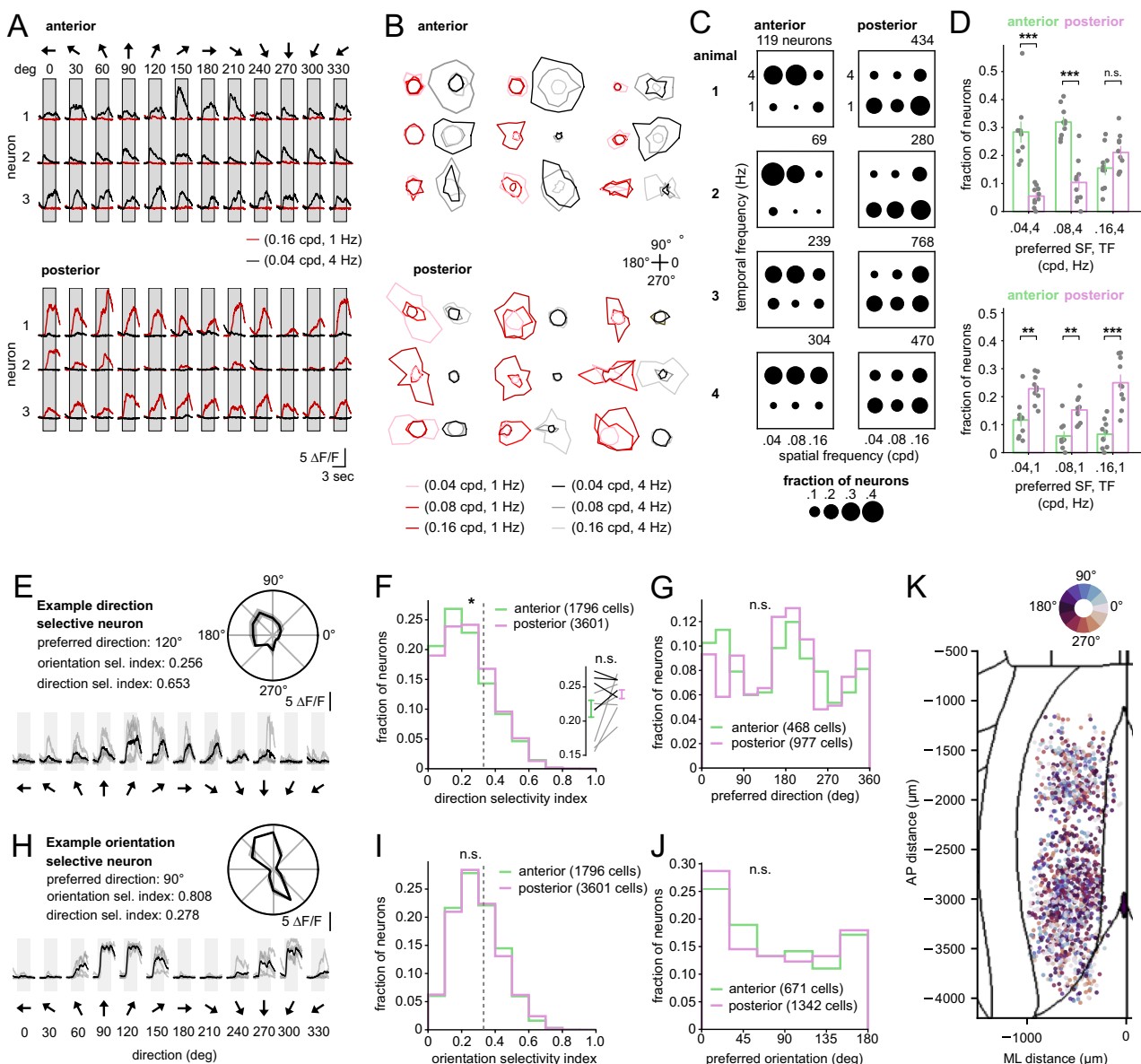

**Fig. 5 | Anterior and posterior RSC visual neurons encode distinct spatial and temporal frequencies. A** Example calcium traces from visually responsive neurons in anterior and posterior RSC. Each column shows a different stimulus direction (30° increments), and each color indicates a distinct spatial and temporal frequency (SF/TF) combination. **B** Visual tuning example with polar plots. Angle represents the stimulus direction; color encodes the combination of spatial and temporal frequencies. **C** Dot plots from four representative animals showing the proportion of visually responsive neurons tuned to each SF/TF combination. Proportions are shown as surface area. **D** Group-level analysis of SF/TF tuning in anterior and posterior RSC. Vertical bars indicate mean ± SEM across sessions. Two-sided Mann–Whitney U test; $p = 1.8\text{e-}4$ (0.04 cpd, 4 Hz); $p = 3.2\text{e-}4$ (0.08 cpd, 4 Hz); $p = 0.18$ (0.16 cpd, 4 Hz); $p = 0.002$ (0.04 cpd, 1 Hz); $p = 0.0017$ (0.08 cpd, 1 Hz); $p = 3.3\text{e-}4$ (0.16 cpd, 1 Hz); $n = 10$ sessions per group (anterior and posterior RSC) from 8 animals. **E** Example direction tuning curve and corresponding polar plot for a direction-selective neuron. Direction selectivity index and orientation selectivity index are shown. **F** Histogram of direction selectivity index (DSI) values for visually

responsive neurons in anterior and posterior RSC. The dotted line (0.33) indicates the classification threshold for direction selectivity. Two-sided Kolmogorov–Smirnov test: $p = 0.012$; 1796 (anterior) vs. 3601 (posterior) neurons ($n = 10$ sessions). No significant difference was observed across sessions (two-sided Mann–Whitney U test: $p = 0.307$; $n = 10$ sessions, gray: Thy1-GCaMP6s; black: CaMKII-GCaMP6). **G** Histogram of preferred directions among direction-selective neurons in RSC. Two-sided Kolmogorov–Smirnov test: $p = 0.064$; 468 vs. 977 neurons ($n = 10$ sessions). **H** Example orientation tuning curve and corresponding polar plot for an orientation-selective neuron. **I** Histogram of orientation selectivity index values for visually responsive neurons. The dotted line (0.33) indicates the classification threshold. Two-sided Kolmogorov-Smirnov test: $p = 0.88$; 1796 vs. 3601 neurons ($n = 10$ sessions). **J** Histogram of preferred orientations for orientation-selective neurons. Two-sided Kolmogorov-Smirnov test: $p = 0.74$; 671 vs. 1342 neurons ($n = 10$ sessions). **K** Dorsal cortical map showing the location of individual visually responsive neurons, color-coded by preferred direction.

4 Hz: $28.3 \pm 3.5\%$ anterior vs. $5.5 \pm 1.1\%$ posterior, $p = 1.8 \times 10^{-4}$; 0.16 cpd at 1 Hz: $6.5 \pm 1.5\%$ anterior vs. $25.0 \pm 2.7\%$ posterior, $p = 3.3 \times 10^{-4}$), and were robust to behavioral modulations (Supplementary Fig. 7). Notably, selectivity for orientation and direction did not differ across subregions (all $p > 0.05$; Fig. 5E–K).

In summary, visual signals are nonuniformly distributed across the RSC. Posterior RSC shows stronger, more reliable responses and preferential sensitivity to slow, high-resolution visual patterns, while anterior RSC is more responsive to fast, low-resolution motion. Notably, we found minimal overlap between neurons encoding visual and

position signals (Supplementary Fig. 8A–D). These findings indicate that visual processing is organized along the anterior–posterior axis.

## Anterior and posterior RSC subregions integrate specific sets of inputs

To determine whether this functional gradient is reflected in long-range anatomical input patterns, we performed brain-wide retrograde tracing. The retrosplenial cortex receives afferents from a broad set of cortical and subcortical regions, including visual cortex, the hippocampal formation (HPF), anterior thalamic nuclei (ATN), posterior parietal cortex (PTLp), and sensorimotor areas[6,7,9,18,20,29,30,42,43]. We therefore asked whether anterior and posterior RSC receive distinct patterns of input from these regions ($n = 4$ animals). To this end, we mapped afferent projections to anterior and posterior RSC at cellular resolution using retrograde AAV tracers (Fig. 6A–C and Supplementary Table 3).

Injections targeted dorsal and ventral subdivisions of RSC at two stereotaxic coordinates: anterior (AP −1.5 mm) and posterior (AP −3.2 mm). In one animal, the posterior injection site was slightly shifted anteriorly (Fig. 6D). After histological processing, labeled neuronal cell bodies were registered to the Allen Mouse Brain Atlas[44] (Fig. 6E). To quantify input distributions, we used two complementary measures: (1) volume-normalized cell density, which accounts for differences in source region size, and (2) fractional input, the proportion of labeled neurons from each region relative to the total input population.

Across the data set, the majority of RSC-projecting neurons originated from the cortex, comprising 80.7 ± 3.2% of inputs to anterior RSC and 71.9 ± 5.9% to posterior RSC (mean ± SEM, $n = 4$ animals; Fig. 6F, G). The hippocampal formation and thalamus (TH) also provided substantial inputs (HPF: 7.6 ± 1.6% and 12.4 ± 2.7%, TH: 6.1 ± 1.4% and 9.9 ± 3.3%, anterior vs. posterior. Fig. 6F–H). Thus, while overall input composition was broadly similar, quantitative differences were observed across regions.

Primary cortical sources included the anterior cingulate area (ACA), motor cortex (MO), somatosensory cortex (SS), posterior parietal cortex, and visual areas (VIS) (Figs. 6H and 7). ACA contributed similarly to both RSC subregions (Fig. 7A, anterior vs. posterior; 17.7 ± 2.3% vs. 19.8 ± 2.4%; mean ± SEM in fraction of inputs; $n = 4$ animals). In contrast, inputs from sensorimotor and parietal cortices were enriched in anterior RSC. MO, SS, and PTLp inputs were more abundant in anterior RSC (Fig. 7B–D, anterior vs. posterior; MO: 23.1 ± 2.7% vs. 7.1 ± 3.8%; $p = 0.125$; SS: 8.7 ± 1% vs. 1.2 ± 0.5%; PTLp: 4.9 ± 0.9% vs. 2.1 ± 0.5%). Posterior RSC received greater input from visual cortices (Fig. 7E; anterior vs. posterior: 7.6 ± 2.1% vs. 23.7 ± 6.9%).

To further characterize differences in visual cortical inputs, we normalized projections from individual visual areas relative to the total VIS-derived input for each RSC subregion. Anterior RSC received proportionally stronger input from higher-order areas such as VISam (anterior vs. posterior: 33.6 ± 2.6% vs. 28.4 ± 6.6%) and VISpor (16.2 ± 7.2% vs. 3.8 ± 1.2%), whereas posterior RSC received more from primary (VISp, 20 ± 4.2% vs. 29.1 ± 5.3%) and posteromedial visual areas (VISpm, 17.2 ± 3.8% vs. 26.0 ± 0.6%). These anatomical biases may contribute to differences in visual tuning profiles across subregions.

To characterize hippocampal inputs (Fig. 8A), we focused on the retrohippocampal region (RHP), which accounted for more than 90% of hippocampal afferents. Most labeled neurons (over 95%) were located in the subiculum (SUB) and entorhinal cortex (ENT). Although posterior RSC received more total RHP input, a greater proportion of the RHP input to anterior RSC originated from ENT (anterior vs. posterior: 51.7 ± 13.8% vs. 27.7 ± 6.7%). Subicular projections showed a clear dorsoventral gradient: dorsal SUB preferentially targeted anterior RSC, while ventral SUB targeted posterior RSC. This topographic organization is consistent with functional specialization within the hippocampus, where dorsal SUB is associated with spatial processing.

Thalamic inputs arose predominantly from the anterior thalamic nuclei, which contributed more than 70% of total thalamic projections. Posterior RSC received a larger overall fraction of ATN input (7.8 ± 2.3% vs. 3.4 ± 0.6%; Fig. 8B). Within ATN, anterior RSC received a higher proportion from the anteromedial nucleus (AM: 52.0 ± 8.2% vs. 33.2 ± 3.3%), whereas posterior RSC received more from the anteroventral nucleus (AV: 46.7 ± 4.1% vs. 30.0 ± 4.6%). Unexpectedly, retro-AAV injections produced minimal labeling in the laterodorsal thalamic nucleus (LD), a major RSC afferent reported previously[29,30]. To determine whether this reflected a lack of connectivity or technical limitations, we performed control injections using cholera toxin subunit B (CTB), which yielded robust LD labeling (Supplementary Fig. 9A, B). Similarly, injecting the same retro-AAV construct into post-subiculum (a region known to receive LD input[45]) failed to label LD neurons (Supplementary Fig. 9C). These controls indicate that viral tropism, rather than absence of LD connectivity, accounts for the underrepresentation of LD inputs in our dataset. Together, these results identify anterior–posterior biases in long-range inputs to RSC that are consistent with functional differences identified in our imaging experiments (Fig. 8C, D).

## Discussion

Combining two-photon calcium imaging with brain-wide retrograde tracing, we identified a systematic anterior–posterior gradient in how visual and position-related signals are represented and integrated within mouse RSC. Anterior subregions exhibit stronger and more reliable position coding and preferential sensitivity to fast, low–spatial-frequency motion, whereas posterior subregions show enhanced responses to slow, high–spatial-frequency stimuli and greater reliance on vision under conditions of structured visual input. Complementing these functional differences, retrograde tracing revealed corresponding anterior–posterior biases in long-range connectivity, with sensorimotor and parietal projections enriched anteriorly and visual cortical inputs enriched posteriorly.

These findings suggest that distinct long-range inputs may differentially support spatial representations across RSC subregions. Prior work demonstrates that secondary motor and posterior parietal cortices provide strong positional signals during navigation[46], consistent with our anatomical results showing denser inputs from motor, somatosensory, and parietal regions to anterior RSC. This connectivity profile provides a plausible substrate for the robust position tuning observed in anterior RSC across both tactile-cued and visually rich environments. Hippocampal–cortical pathways also contribute to RSC position coding[20,47], and our tracing indicates that both subiculum and entorhinal cortex project broadly to RSC. Notably, dorsal subiculum, associated with geometric and spatial encoding[48–50], preferentially targeted anterior RSC, whereas ventral subiculum, associated with contextual and affective signals[51], projected more strongly to posterior RSC. These gradients are consistent with the possibility that anterior and posterior RSC differentially sample components of hippocampal output, potentially contributing to their distinct roles in navigation.

Consistent with these connectivity differences, posterior RSC showed weaker position tuning in the tactile-cued environment but exhibited enhanced tuning in visually rich VR conditions, suggesting a greater influence of structured visual input. Together, these findings support a model in which position coding in RSC emerges from the integration of multiple sensory modalities, with the relative contribution of each modality differing systematically across subregions in a manner consistent with their long-range connectivity profiles. The functional segregation may enable the RSC to integrate tactile, motor, and self-motion cues with visual landmarks to generate stable spatial representations across navigational contexts.

In addition to differences in position coding, we observed a corresponding anterior–posterior gradient in visual processing. Consistent with previous reports[28,34], posterior RSC showed stronger visual

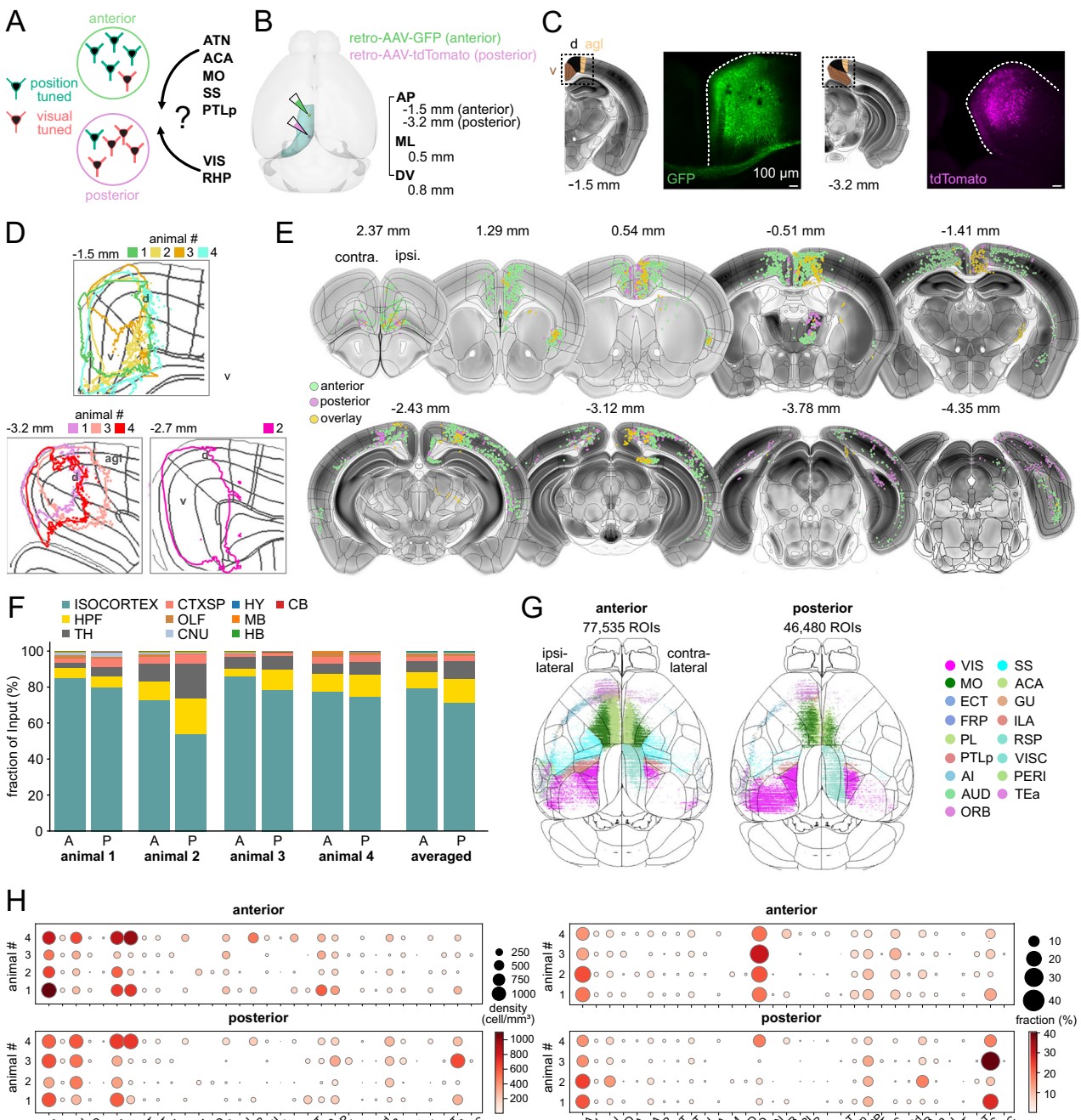

**Fig. 6 | Brain-wide retrograde mapping reveals distinct long-range input sources to anterior and posterior RSC. A** Schematic of the experimental design for mapping differential afferent connectivity to anterior and posterior RSC using retrograde viral tracing. **B** Retrograde AAV constructs encoding fluorescent proteins were injected into anterior and posterior RSC in wild-type mice to label upstream input neurons. Image created using the brainrender open-source Python library. **C** Fluorescent expression at the injection site three weeks post-injection, shown in coronal brain sections. Representative images from 4 animals. Scale bar, 100 μm. **D** Normalized 90th percentile fluorescence intensity around the injection site for each animal (*n* = 4 animals). The representative image in (**C**) corresponds to animal 1. **E** Example of serial coronal sections showing identified retrogradely labeled neurons. Coordinates relative to Bregma are indicated above each section. **F** Distribution of labeled neurons across major brain regions for four animals. HPF hippocampal formation, TH thalamus, CTXsp cortical subplate, OLF olfactory areas, CNU cerebral nuclei, HY hypothalamus, MB midbrain, HB hindbrain, CB

cerebellum. **G** Dorsal view of the cortex showing the distribution of retrogradely labeled neurons across the isocortex. **H** Brain-wide input distribution by region. Left: volume-normalized cell density. Right: proportion of total labeled input neurons. Dot size and color indicate relative input strength. ACA anterior cingulate area, AI agranular insular area, ATN anterior group of the dorsal thalamus, AUD auditory areas, VIS visual areas, CA ammon's horn, CLA claustrum, DP dorsal peduncular area, ECT ectorhinal area, ENT entorhinal area, GU gustatory areas, ILA infralimbic area, ILM intralaminar nuclei of the dorsal thalamus, MED medial group of the dorsal thalamus, MO somatomotor areas, MTN midline group of the dorsal thalamus, ORB orbital area, PERI perirhinal area, PIR piriform area, PL prelimbic area, POST postsubiculum, PTLp posterior parietal association areas, RSP retrosplenial area, RT reticular nucleus of the thalamus, SS somatosensory areas, STRd striatum dorsal region, SUB subiculum, TEa temporal association areas, VENT ventral group of the dorsal thalamus, VISC visceral area.

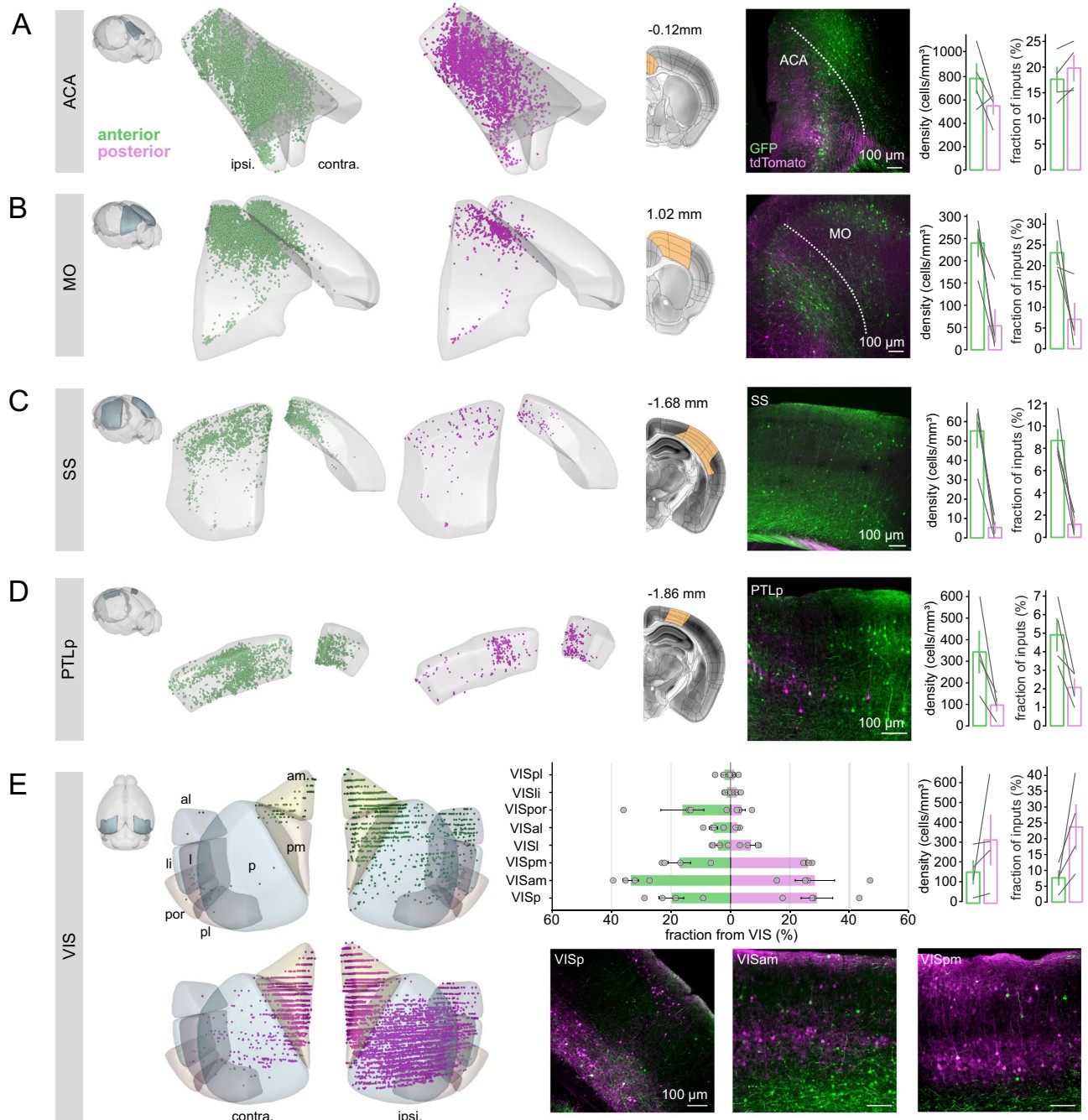

**Fig. 7 | Anterior and posterior RSC neurons receive distinct afferent inputs from sensorimotor and visual areas. A–D** Retrogradely labeled neurons in ACA, MO, SS, and PTLp. From left to right for each region: (i) 3D reconstruction of the brain area showing labeled neurons from a representative animal, (ii) confocal image and corresponding coronal section (scale bar, 100 μm), and (iii) quantification of volume-normalized density and proportion of labeled neurons across animals, vertical bars with each connected line indicating an individual animal

(mean ± SEM, $n = 4$ animals). **E** Similar layout as in (**A–D**) but showing retrogradely labeled neurons in visual areas (VIS). Horizontal bar plots show the fraction of inputs from each VIS subregion, normalized by the total number of labeled neurons across all VIS subregions (mean ± SEM, $n = 4$ animals). Scale bar, 100 μm. VISal anterolateral visual area, VISam anteromedial visual area, VISl lateral visual area, VISpm posteromedial visual area, VISli laterointermediate area, VISp primary visual area, VISpor postrhinal area, VISpl posterolateral visual area.

responses than anterior RSC. Beyond this difference in response strength, our data further indicated a systematic gradient in visual tuning, indicating that the two subregions emphasize different visual features during navigation.

These gradients parallel tuning differences across visual cortical areas[39–41] and are consistent with the projection patterns we observed− posterior RSC received stronger input from primary (VISp) and

posteromedial visual areas (VISpm), while anterior RSC was preferentially innervated by higher-order regions such as VISam and VISpor. This correspondence raises the possibility that visual response properties in RSC are shaped, at least in part, by the structure of its long-range cortical inputs, although intrinsic circuit mechanisms may also contribute. Our widefield retinotopic mapping further revealed coarse retinotopic organization, with posterior RSC responding more

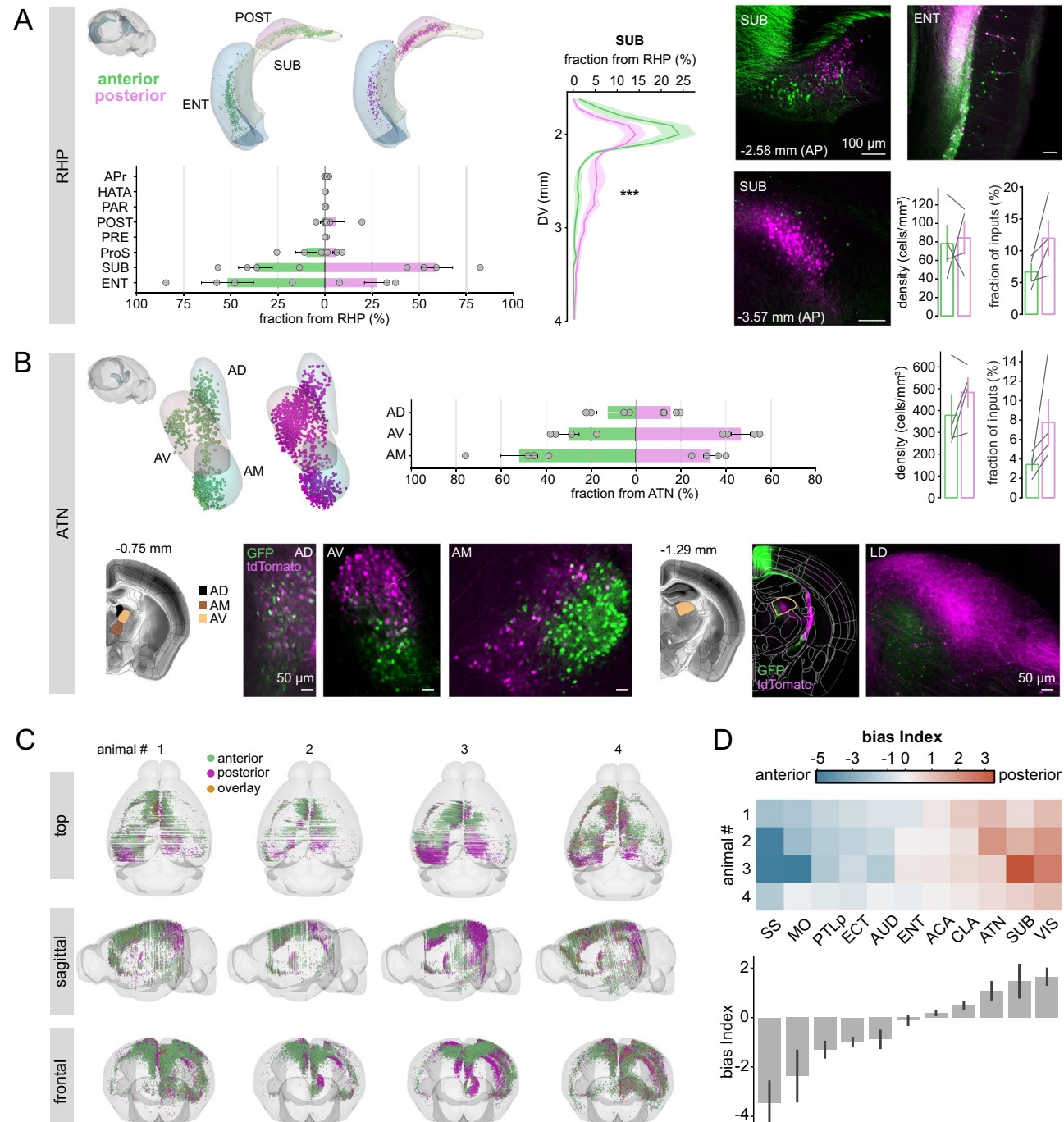

**Fig. 8 | Anterior and posterior RSC neurons receive distinct afferent inputs from parahippocampal and anterior thalamic areas. A** Retrogradely labeled neurons in the retrohippocampal area (RHP). From left to right: (i) 3D reconstruction of the brain area showing labeled neurons from a representative animal, and horizontal bar plots show the fraction of inputs from each RHP subregion, normalized by the total number of labeled neurons across all RHP subregions (mean ± SEM, *n* = 4 animals), (ii) line plots show the topographical projection from the dorso−ventral axis of SUB to anterior versus posterior RSC. Kolmogorov-Smirnov test; *p* = 1.7e-6; 2466 (anterior) vs. 4046 (posterior) rois (*n* = 4 animals), (iii) confocal image (scale bar, 100 μm), and (iv) quantification of volume-normalized density and proportion of labeled neurons across animals, vertical bars with each connected line indicates an individual animal (mean ± SEM, *n* = 4 animals). ENT entorhinal area, SUB subiculum, POST postsubiculum, APr area

prostriata, PRE presubiculum, HATA hippocampo-amygdalar transition area, PAR Parasubiculum. **B** Retrogradely labeled neurons in anterior dorsal thalamic nuclei (ATN). Horizontal bar plots show the fraction of inputs from each ATN subregion, normalized by the total number of labeled neurons across all ATN subregions (mean ± SEM, *n* = 4 animals). Scale bar, 50 μm. AM anteromedial nucleus, AV anteroventral nucleus, AD anterodorsal nucleus, LD lateral dorsal nucleus. **C** 3D reconstruction of the input from all regions across animals (*n* = 4 animals). **D** Top: Bias index for the selected brain areas for each animal. Bottom: Values are shown as mean ± SEM (*n* = 4 animals). Note on normalization: The bias index reflects the absolute input contribution from each brain region to the entire RSC input pool and thus captures the global distribution of inputs. This differs from the horizontal bar plots in (**A**, **B**), which display the relative proportion of input from each sub-region within a given anatomical group.

strongly to stimuli in the nasal upper visual field and anterior RSC to stimuli in the nasal lower field (Supplementary Fig. 5), broadly consistent with cellular studies[28]. Differences in stimulus design and measurement scale may account for the relatively weak azimuthal gradients observed here compared with earlier work. Together, these findings link anterior–posterior differences in visual tuning to systematic variation in long-range cortical inputs, consistent with hierarchical organization in the visual system.

Because our two-photon imaging targeted superficial layers, ventral RSC (RSCv) comprised only a small fraction of recorded neurons (10–20%), and it is therefore unlikely that differences in dorsal–ventral sampling account for the anterior–posterior functional distinctions reported here. Thus, the observed gradient cannot be readily explained by subdivision sampling biases alone. Our retrograde AAV injections labeled neurons projecting to both dorsal and ventral subdivisions of RSC, indicating that long-range inputs were broadly sampled across the cortical depth. Despite modest variability in injection placement, the resulting connectivity patterns were consistent across animals, consistent with the robustness of the anterior–posterior input gradient.

Whether this gradient reflects discrete modules or a continuous functional organization remains unresolved. Although anatomical and transcriptomic studies have identified heterogeneity within RSC[30,52], they have not yet established sharply defined anterior and posterior modules. Future studies employing spatially resolved molecular profiling and causal manipulations (e.g., optogenetics, chemogenetics, or targeted inactivation) will be essential to determine whether functional specialization along the anterior–posterior axis reflects intrinsic circuit differences or graded variation in afferent input.

Finally, our study was limited to head-fixed navigational tasks, which limit access to vestibular and full-body motor cues contributing to egocentric spatial representations. As a result, our findings primarily reflect how visual, tactile, and self-motion cues are integrated under controlled sensory conditions. The RSC integrates allocentric and egocentric information from hippocampal, thalamic, and visual systems[18,21,45,53–55] and participates in processing egocentric boundary and landmark signals[20,21]. It therefore remains possible that freely moving conditions engage additional reference-frame transformations that are unevenly distributed along the anterior–posterior axis. Testing how vestibular and body-derived cues are incorporated into these subregions will help determine whether the functional gradient described here generalizes to naturalistic navigation.

Importantly, our results demonstrate that even under these constrained conditions, position and visual coding are differentially organized along the anterior–posterior axis of RSC, suggesting that this organizational principle is robust to task context. This raises the possibility that egocentric representations, like allocentric ones, may also vary systematically across RSC subregions. Addressing this question will require freely moving paradigms and targeted circuit interventions to examine how different sensory reference frames are combined across RSC subregions during naturalistic navigation.

Together, our results characterize how anatomically distinct RSC subregions integrate sensory information to support visuospatial computations and guide navigation. By identifying distinct roles for anterior and posterior RSC, this work provides a basis for future studies probing how these circuits operate across behavioral contexts and sensory environments.

## Methods
### Animals
All experimental procedures were approved by the Animal Ethics Committee of KU Leuven. For chronic cellular calcium imaging (see Supplementary Tables 1 and 2), we used either C57BL/6J-Tg(Thy1-GCaMP6s)GP4.12Dkim/J mice[56] or CaMKII-tTA x TRE-GCaMP6s (line G6s2) mice[57]. Wild-type C57BL/6J mice were utilized for retrograde

tracing experiments (Supplementary Table 3). All animals were between 3 and 8 months old at the time of the experiments. Mice were single-housed in an enriched environment consisting of cotton bedding, wooden blocks, and a running wheel, under a 12 h light/dark cycle, with room temperature maintained at 19–21 °C and relative humidity ranging from 30 to 70%.

### Resources
Non-animal resources used for this study are listed in Supplementary Table 4.

### Cranial window surgery
To enable chronic cellular calcium imaging, mice were anesthetized with isoflurane (2.5–3% for induction, 1–1.25% for surgery). A custom-made titanium frame was attached to the skull, and a cranial glass window (4 or 5 mm, 2–3 layers) was implanted over the left hemisphere covering the RSC[58]. Postoperative care included administration of Buprenex (0.6 mg/kg) and Cefazolin (15 mg/kg) every 12 h for two days to manage pain and prevent infection.

### Habituation and behavioral training
Following recovery, mice were gradually habituated to head restraint while positioned on a linear treadmill. To motivate locomotion during the behavioral task, animals were placed on a water restriction schedule, receiving no less than 1 mL of water per day. Body weight was monitored daily to ensure it remained above 80% of their free-feeding weight.

Habituation progressed from short initial sessions to durations of up to 1 h per day. During this period, animals were introduced to the experimental setup without tactile cues. A small water reward (~2–3 μL) was delivered at a fixed location following each completed lap. Additional rewards were occasionally provided to encourage task-related behaviors, such as licking and running. Once animals exhibited stable locomotion (>5 cm/s), they were transitioned to either the tactile treadmill task or the virtual reality task.

### Tactile treadmill task
Two tactile cues (gray sponge or foam patches) were affixed to the treadmill belt at 50 cm and 100 cm positions. A small water reward (~2–3 μL) was delivered at a fixed location following each completed lap.

### Virtual reality task
To examine position-related activity in a visually landmark-rich environment, locomotion was synchronized in real-time with presentation of a virtual corridor[15]. The environment was displayed on a single monitor (Samsung 2233RZ; 1680 × 1050 resolution; 60 Hz refresh rate) covering ~120° of the frontal visual field, spanning the binocular zone of both eyes. The virtual track was 210 cm in length and consisted of two walls covered with randomized dot textures and a vertical grating landmark positioned at 100 cm. Upon reaching the end of the corridor, the visual scene terminated, and the animal was teleported to a reward zone where it received ~2–3 μL water reward over 3 s to begin the subsequent trial. To minimize transient visual responses, all blocks were presented at the same mean luminance. In the open-loop condition (decoupled from locomotion), the virtual environment moved passively at ~6 cm/s.

### Two-photon calcium imaging
During imaging experiments, eye movements and facial expressions were simultaneously recorded using an integrated tracking system with AVT Prosilica GC660 and Mako G-030 cameras, respectively.

Cellular calcium imaging was conducted using a resonant scanner-based two-photon microscope (Neurolabware). In a subset of the experiments, an electrically tunable lens (ETL) was used to

sequentially image across four optical planes, spanning a depth range of 60–300 μm, with an inter-plane spacing of 60–100 μm (see Supplementary Table 1). For single-plane imaging, data were acquired at ~30 frames per second, while ETL-based volumetric imaging achieved an effective frame rate of ~7.5 Hz per plane. The genetically encoded calcium indicator GCaMP6 was excited at 920 nm using a MaiTai DeepSee laser (Spectra Physics) through a 16× water-immersion objective (NA = 0.8, Nikon). The emitted green light was filtered (510/84 nm, Semrock) and detected using a GaAsP photomultiplier tube (PMT, Hamamatsu). Laser power output at the objective ranged from 20 to 100 mW, increasing with imaging depth. Images were acquired at a resolution of 794 × 527 pixels, corresponding to a physical area of 892 × 667 μm. To minimize contamination from ambient light, blackout material was used to shield the optical path from the visual display.

### Single-photon calcium imaging

Widefield single-photon calcium imaging was used to map the retinotopic organization and delineate the boundaries of dorsal cortical areas. Excitation light was provided by a 470 nm blue LED (Thorlabs), and fluorescence emission was collected through a 510/84 nm green emission filter (Semrock) using a CCD camera (PCO Edge 3.1). Imaging was acquired at 10 frames per second using a 2× objective lens (NA = 0.10, Thorlabs), yielding an image resolution of 1024 × 768 pixels across the field of view.

### Visual stimulation

Visual stimuli were presented on a calibrated 22-inch LCD monitor (Samsung 2233RZ, 1680 × 1050 resolution, 60 Hz refresh rate), positioned 18 cm from the animal's right eye. The screen covered 120° × 80° of the right visual field (0–120° azimuth; ±40° elevation). Custom software built in PsychoPy[59] was used to control stimulus presentation and synchronize with imaging acquisition.

For retinotopic mapping, we presented a circular patch of flickering noise rotating counterclockwise along an elliptical trajectory centered on the display (Supplementary Movie 1). Each trial consisted of four rotations, followed by a 6 s inter-trial interval, repeated across 30 trials.

For cellular calcium imaging, full-screen square wave drifting gratings of 12 directions and 6 combinations of spatial frequencies (0.04, 0.08, and 0.16 cpd) and temporal frequencies (1 and 4 Hz) were presented, covering the range of visual tuning observed in the visual cortex of awake mice[38,39]. Stimuli were presented in pseudorandomized order at fixed 5 s intervals with 3 s of visual stimulation interleaved with 2 s of gray screen (54–80 cd/m², from center to edge of display). In dark-condition experiments, the monitor and all external light sources (e.g., for pupil or facial tracking) were turned off.

### Viral vectors and CTB injections

To label neurons projecting to anterior and posterior RSC, retrograde AAV viral vectors were injected through a burr-hole craniotomy using a glass micropipette (30 μm tip diameter, Drummond Scientific Company, Pulling system from Sutter Instrument P-2000). Two combinations of viral vectors with identical promoters were used: AAVrg-CAG-tdTomato with AAVrg-CAG-GFP, or AAVrg-hSyn-mCherry with AAVrg-hSyn-GFP (titer: ≥7×10¹² vg/mL, Supplementary Tables 3 and 4). Injections were targeted using stereotaxic coordinates relative to Bregma. For the anterior RSC, the coordinates were anteroposterior (AP) −1.5 mm, mediolateral (ML) 0.5 mm, and dorsoventral (DV) −0.8 mm. For the posterior RSC, the coordinates were AP −3.2 mm, ML 0.5 mm, and DV −0.8 mm. Each injection was made approximately 0.5 mm below the cortical surface. A volume of 150–200 nL viral solution was infused at each site, using a Nanoject II injector that delivered ~13 nL per pulse at 20-s intervals.

To confirm thalamic input to RSC from the laterodorsal (LD) nucleus, we performed additional retrograde tracing using a non-viral method. A 0.5% solution of cholera toxin subunit B (C34775, ThermoFisher) was injected into either anterior or posterior RSC at the exact stereotaxic coordinates described above[60].

### Histology

Three weeks following viral injection (or 2 weeks after CTB injection), mice were deeply anesthetized with an intraperitoneal injection of ketamine and xylazine (100 mg/kg and 10 mg/kg, respectively), and transcardially perfused first with phosphate-buffered saline (PBS), followed by 4% paraformaldehyde (PFA). Brains were extracted and cut into 100 μm-thick coronal sections using either a vibratome (VT1000 S, Leica) or a cryostat (CM3050 S, Leica). Sections were mounted using Vectashield containing the nuclear stain DAPI (4',6-diamidino-2-phenylindole; Vector Laboratories). Confocal imaging was performed with a Zeiss LSM 900 microscope or Nikon Ti2 inverted microscope equipped with a 10× Plan-APOCHROMAT objective (NA = 0.45). Images were acquired using a 0.7× zoom setting with approximately 12 by 7 tile scans and a 15% overlap between tiles at a resolution of either 1.25 (1024 × 1024) or 2.5 μm (512 × 512) per pixel. Z-stacks were acquired with a step size of 5–7.5 μm and covered at least 80 μm in depth.

### Quantification and statistical analysis

All data analyses were performed using custom scripts written in Python and MATLAB.

### Widefield calcium imaging data

For widefield calcium imaging, fluorescence signals were first normalized to the pre-stimulation baseline. A min-max filter was applied to enhance spatial and temporal contrast (Supplementary Movie 1). Trial-averaged data were analyzed using a discrete Fourier transform along the temporal axis to extract pixel-wise frequency components, with amplitude representing response strength and phase indicating response timing. These components were visualized using an HSV colormap based on normalized amplitude and phase. Registration to the Allen Common Coordinate Framework (CCF) dorsal cortex was achieved by matching retinotopic maps in primary and higher visual areas[44].

### Cellular calcium imaging data

Cellular calcium imaging data were acquired using ScanBox and processed with Suite2P for motion registration, ROI detection, and calcium signal extraction[61]. The extracted neuronal activity included the raw fluorescence signal $F_{cell}$ and neuropil fluorescence signal $F_{np}$. For each ROI, the raw somatic fluorescence signal was corrected by subtracting surrounding neuropil activity using the formula:

$$F_{corr} = F_{cell} - 0.7 \times F_{np}$$

The baseline fluorescence was estimated using a min/max filter with 60 s kernel size, applied to $F_{corr}$, followed by Gaussian smoothing ($\sigma$ = 10 time bins). Spike deconvolution was performed using the OASIS algorithm, which models the calcium trace as the convolution of spike events with an exponential decay kernel. A fixed decay constant (1.5 s) appropriate for GCaMP6s was used[62].

### Selection of active neurons

To identify neurons actively engaged during the behavioral task, we applied separate criteria for movement-related and visually evoked activity. For task-related activity, lap reliability was calculated as the proportion of running laps in which a neuron exhibited significant calcium transients. Transients were considered significant if the fluorescence signal exceeded the baseline $\Delta F/F_0$ by more than three

times the standard deviation. This criterion ensured the inclusion of neurons that consistently responded across repeated laps.

For assessing visual responsiveness, trial-averaged $\Delta F/F_0$ traces for each stimulus condition were baseline-corrected by subtracting the median pre-stimulus signal, calculated over the five frames preceding stimulus onset. To quantify the consistency of visually evoked responses, a visual reliability index ($r$) was computed as the 75th percentile of cross-trial Pearson correlation coefficients of the de-randomized response time courses. A neuron was classified as visually responsive if $r > 0.3$[39].

A neuron was considered active during the task if it exhibited either lap reliability greater than 0.3 (i.e., significant activity in at least 30% of laps) or visual reliability greater than 0.3.

### Quantification of speed score
The speed score was calculated by the Pearson product-moment correlation between the cell's instantaneous firing rate and the mouse's instantaneous running speed, which scales from −1 to 1[63].

### Population decoding of position
We employed a Bayesian algorithm[64,65] to estimate the probability of position given the $\Delta F/F_0$ of a fixed number of randomly selected neurons. Bayesian decoding was performed during running epochs, defined as a consecutive sequence of frames with forward movement lasting at least 1 s, with a minimum speed of 5 cm/s. Consecutive epochs separated by less than 0.5 s were merged[66].

To compare the decoding error between the anterior and posterior retrosplenial cortex, odd trials were used for model training, while even-numbered trials were reserved for testing. To assess decoding error in darkness, 80% of the total trials from the light-on session were used for training, and the remaining trials were reserved for testing using a 5-fold cross-validation approach. The Bayesian decoding equation is given by:

$$P(\text{pos}|a_{\text{all}}) = C \left( \prod_{i=1}^{N} f_i(\text{pos})^{a_i} \right) e^{-\tau \sum_{i=1}^{N} f_i(\text{pos})}$$

where $a_{\text{all}}$ represents the activity ($\Delta F/F_0$) of the neurons, and $C$ is the normalization constant ensuring that the posterior sums to 1. The parameter $\tau$ is the time bin size, which depends on the imaging sampling rate. The function $f_i(\text{pos})^{a_i}$ represents the position-binned activity (with 1.5 cm per bin), and $N$ is the number of neurons. The decoded position for each time bin was determined as the position with the maximum posterior probability. The decoding error was computed as the absolute difference between the actual position and the decoded position. Chance-level decoding was set at 50 spatial bins (75 cm).

### Detection of neurons with position activity
Position-tuned neurons were identified using a lower-bound threshold computed from deconvolved calcium data[67]. The linear track was divided into 100 position bins (1.5 cm per bin), and the occupancy-normalized activity was smoothed using a Gaussian window (SD = 3 position bins) for each neuron. To determine statistical significance, neuronal activity was circularly shifted by a random time between 20 s and the total session duration minus 20 s, repeated 200 times to generate a shuffled distribution. A neuron was classified as position-tuned if the lower bound of its actual activity (mean-SEM) in any position bin across trials exceeded the 97.5th percentile of the shuffled distribution.

Position-tuned neurons were further examined by quantifying their position-tuning properties. The position tuning field width was calculated from the number of consecutive position bins in which the mean activity exceeded 30% of the difference between peak and baseline activity (below 15 cm or above 120 cm were discarded). The neuron was considered a position tuned if it met the following criteria: (1) Position tuning field width was a continuous region spanning at least 15 cm but no more than 120 cm. (2) At least one position tuning field was presented in more than one-third of all trials.

### Quantification of position-related activity
Spatial information (SI) was calculated using the following formula[68]:

$$\text{SI} = \sum_{i=1}^{N} p_i \frac{f_i}{f} \log_2 \left( \frac{f_i}{f} \right)$$

where $p_i$ is the occupancy probability (the fraction of the time spent in the -th position bin), $f_i$ is the occupancy-normalized deconvolved calcium activity (summed activity divided by the total time spent in the -th position bin), and $f$ is the overall mean activity.

To examine activity reliability across trials (whether calcium transients occurred at consistent locations across trials), we computed trial-to-trial correlation as the median of the pairwise Pearson correlation coefficients across trials using deconvolved calcium activity.

### Characterization of direction and orientation visual selectivity
Orientation and direction tuning were quantified using the orientation selectivity index (OSI) and directional selectivity index (DSI), respectively:

$$\text{OSI} = \frac{R_p - R_{\text{ortho}}}{R_p + R_{\text{ortho}}} \quad \text{DSI} = \frac{R_p - R_{\text{null}}}{R_p + R_{\text{null}}}$$

where $R_p$ is the response to the preferred orientation/direction, $R_{\text{ortho}}$ is the response to the orthogonal orientation, $R_{\text{null}}$ is the response to the 180° direction from the preferred direction. Neurons with OSI > 0.33 or DSI > 0.33 were considered orientation and direction-selective, respectively[69].

### Quantification of retrogradely labeled cells
Labeled neurons were manually selected in different channels separately. Registration was performed using the DAPI channel across all brain slices to the Allen Brain Atlas[70]. After applying a transformation matrix to the brain slice, each ROI was registered to a specific brain region based on the mouse brain atlas. ROIs were classified according to the hierarchy structure tree provided by the Allen Brain Institute[44]. A 3D whole-brain reconstruction with labeled ROIs was generated based on the model adapted from brainrender[71].

To assess the relative input distribution between posterior and anterior subregions of the retrosplenial cortex (RSC), we computed a bias index defined as the log-ratio of the fractional input to posterior RSC versus anterior RSC:

$$\text{Bias Index} = \log_2 \left( \frac{f_{\text{posterior}}}{f_{\text{anterior}}} \right)$$

where $f_{\text{posterior}}$ and $f_{\text{anterior}}$ represent the fraction of input to the posterior and anterior RSC, respectively. Positive values indicate a relative bias toward posterior RSC, whereas negative values indicate a bias toward anterior RSC.

### Quantification of fluorescence density
Fluorescence density was quantified from histological sections at the viral injection sites following standardized preprocessing and normalization steps. Images were smoothed to reduce high-frequency noise and normalized using intensity percentiles to correct for differences in overall brightness across sections. Background signal was estimated from non-labeled regions and subtracted, after which fluorescence values were rescaled to a standard intensity range. Each

processed section was registered to the Allen Common Coordinate Framework (CCF), and the 90th percentile fluorescence density was visualized as contour plots.

## Statistical testing

Statistical analyses were performed to assess differences in neuronal properties between anterior and posterior RSC across animals. To compare scalar summary measures, we used the two-sided Mann–Whitney U test, a non-parametric alternative to the t-test that does not require normality assumptions. For population-level comparisons, we used the two-sample Kolmogorov–Smirnov test, which evaluates differences between distributions. Statistical significance levels are reported as follows: $p < 0.05$ (*), $p < 0.01$ (**), $p < 0.001$ (***).

## Reporting summary

Further information on research design is available in the Nature Portfolio Reporting Summary linked to this article.

## Data availability

All two-photon imaging datasets (Suite2p-processed) and retrograde-labeling data (CSV format) have been deposited in Zenodo and are publicly available (https://doi.org/10.5281/zenodo.17639283). Any additional data supporting the findings of this study are available from the corresponding author upon request.

## Code availability

All custom code used in this study is openly available on GitHub (https://github.com/ytsimon2004/rscvp) and archived in Zenodo (https://doi.org/10.5281/zenodo.18234118). The public repository includes detailed documentation and Google Colab notebooks to facilitate streamlined data download and full reproduction of the analyses. Further information required to reproduce or reanalyze the results is available from the corresponding author upon request.

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

## Acknowledgements

We thank Ben Vermaercke, Adrien Philippon, and Ta-Shun Su for technical assistance. YT.W. was supported by a scholarship from the Taiwanese Ministry of Education (MOE)–KU Leuven. J.C. acknowledges support from FWO (Fellowship 1226320N). V.B. acknowledges support from FWO (grants G0C1220N, G083J24N, and G056725N). V.B. and F.K. acknowledge support from FWO (grant G077321N). We are also grateful to the members of the Bonin and Kloosterman labs for valuable discussions and feedback.

## Author contributions

Conceptualization: YT.W., J.C., and V.B.; Methodology: YT.W. and V.B.; Software: YT.W.; Investigation: YT.W., V.B., and F.K.; Data Analysis: YT.W.; Visualization: YT.W. Writing—original draft: YT.W.; Writing—review & editing: YT.W., V.B., and F.K.; Funding acquisition: J.C., V.B., and F.K.; Supervision: V.B. and F.K.

## Competing interests

The authors declare no competing interests
