## [Transparent Peer Review file · Nature Communications]

Anterior and posterior retrosplenial cortex form complementary visuospatial circuits in the mouse

Corresponding Author: Professor Vincent Bonin

Version 0:

Reviewer comments:

Reviewer #1

(Remarks to the Author)

This manuscript investigates multimodal information encoding and long-range input connectivity along the rostro-caudal axis of retrosplenial cortex (RSC), revealing gradients in spatial and visual processing and connectivity. The topic is timely, as systematic subregion analyses in RSC are rare, and the data could substantially advance our understanding of RSC's role in sensory, spatial, and mnemonic functions. However, several issues must be addressed to support the authors' conclusions for publication.

Major comments:

1- Rostro-caudal spatial gradient vs. sensory drive: The observed position-related activity may reflect multimodal integration of somatosensory, visual, and self-motion cues, rather than pure "spatial" coding. Considering the use of tactile cues as landmark on the treadmill, the stronger rostral position-related signals may arise from stronger somatosensory inputs to the rostral, as compared to caudal, RSC rather than a generalised higher spatial content. Indeed, the viral tracing data in Figure 6H suggest stronger somatosensory cortical inputs to rostral RSC. To conclude that the observed difference reflects spatial coding per se, the study could test position-related signals with non-tactile landmarks (visual cues and running alone) to show that the gradient persists, or temper the conclusion to acknowledge that stronger rostral coding could arise from greater somatosensory and motor inputs, as supported by the tracing data.

2- Underestimated thalamic input: The use of retrograde viral tracing and whole-brain quantification of input connectivity provides a rich and informative dataset that complements the functional data. However, there is a major concern regarding thalamic connections: the results show minimal laterodorsal (LD) nucleus input, which contradicts extensive literature identifying the LD as one of the key thalamic afferents to the RSC. Since this discrepancy may stem from viral tropism, I suggest validating the results with an alternative viral or structural retrograde tracing method before drawing firm conclusions about connectivity.

3- Sampling across RSC layers and subregions: Since a major finding of the study is rostro-caudal gradient in spatial and visual information processing, it is important to demonstrate similar sampling across the subregions – granular (gRSC) vs dysgranular (dRSC) areas – and cortical layers (2/3 vs 5) along the rostro-caudal axis. For instance, rostro-caudal differences in cortical layer depth and dRSC/gRSC boundaries may lead to oversampling from gRSC in more rostral regions or oversampling of layer 5 in more caudal regions (300 µm depth is well within layer 5 in caudal RSC). This issue applies to both functional and anatomical data, and can be addressed by showing the laminar and regional distribution of imaged neurons and viral injection spread for both rostral and caudal sections to demonstrate similar sampling.

4- Subicular and ENT input, position coding, and Figure 7G-H: The discussion section suggests that the position-related signal in RSC is hippocampal in origin. If so, one would predict that rostral RSC's stronger position coding should coincide with stronger input from subiculum (the hippocampus's primary output). Instead, Figure 8 shows a posterior bias in subicular input for 3/4 animals (positive bias index). Similarly, although the text states that entorhinal cortex (ENT) input is stronger to anterior RSC, the mean ENT bias index is near zero, and individual animals have both positive and negative biases. It is unclear how the indices in Figure 7H align with the stacked bar chart in Figure 7G, since the panel captions lack detail.

The manuscript also describes a dorsal–ventral gradient in subicular projections to anterior vs. posterior RSC, yet no per-animal or group-level quantification is provided. If Figure 7G illustrates this gradient, please explain the panel in the caption to define all colours and results clearly (it's very difficult to understand this panel as is), and include statistics (mean \pm SEM, test values, p-values) to support any regional differences.

Overall, the claim that rostro-caudal differences in position coding are explained by parahippocampal connectivity is not fully supported, because i) the gradient may reflect sensorimotor integration rather than hippocampal-dependent spatial coding (see comment 1); ii) it is unclear whether subiculum itself exhibits a comparable dorsal–ventral gradient in position coding (please cite any relevant studies); iii) a significant difference in dorsal vs. ventral subicular input to RSC has not been convincingly demonstrated.

Additional comments:

- Powell et al. (2020) demonstrated a rostro-caudal gradient in visual responses in RSC. Please discuss and contrast their findings in the discussion section on visual properties. The text currently implies that no prior study has explored this question in the context of spatial and visual processing and that prior knowledge about rostro-caudal organization was limited to object recognition and contextual fear (lines 421–427).
- Page 3, line 91: The description of imaged areas is unclear and does not match Figure 1C. The text describes imaging “in RSPd within the agranular layer of the RSC at depths of 60–360 μ m below the pial surface.” Dysgranular and agranular RSC are two separate but adjacent subregions. Additionally, Figure 1C shows the FOV covering both dysgranular (RSCd) and part of granular RSC (referred to as RSCv in this manuscript). Finally, based on the Allen Brain Atlas, this imaging depth covers both layer 2/3 and layer 5 (especially in caudal RSC, as explained above). Please clarify the imaging location with details about layers and subregions in both rostral and caudal RSC and resolve the inconsistency between the text and Figure 1C.
- For all statistical analyses, neurons across all animals are pooled and treated as independent replicates. This inflates N and raises the risk of type I errors (small effects may appear significant even if between-animal variance is large). I recommend either collapsing each animal's anterior and posterior responses into per-animal summary statistics and testing across animals, or fitting a mixed-effects model with animal as a random effect.
- Two different transgenic mouse lines are used in this study. What was the rationale, and are the results consistent between these two lines?
- Page 12, lines 306-309: Please include SEM values.
- Page 13, lines 318 -322, lines 330- 335, lines 339 – 342: Statistics are missing in all these sections that conclude differences between anterior and posterior RSC connectivity.
- Page 13, line 328: revise “anterior dorsal” to “anterior” thalamic nuclei (ATN)

Reviewer #2

(Remarks to the Author)

In this manuscript, Wei et al. investigate differences in connectivity and functional responses between anterior and posterior dysgranular retrosplenial cortex (dRSC). They do some convincing two-photon imaging experiments to show that anterior and posterior dRSC show differences in visual and allocentric spatial tuning. They also do some nice histology using retrograde viral tracing to quantify differences in connectivity. These findings, though perhaps of somewhat specialized interest, would make a nice contribution to the field. However, I do have a few concerns about specific analyses and their scholarship, as described below:

Major:

1. The scholarship is not very thorough when discussing differences in anterior and posterior connectivity. For example, in the introduction they cite papers that have nothing to do with anterior-posterior gradients of visual/hippocampal input (e.g., Niell and Stryker, 2008, Goldbach et al., 2021, etc.), while skimming over or missing papers of very direct relevance. For example, Yamawaki et al., J Neuro, 2016 and Powell et al, Cereb Cortex, 2020 (both cited elsewhere in the manuscript, but not here) both offer direct evidence of anatomical and functional gradients along anterior-posterior RSC. The authors also completely missed Franco & Goard, Science Advances, 2021, which show both anatomical and functional gradients in anterior-posterior RSC using similar techniques.

As a result, the authors present certain results—such as stronger visual tuning in posterior dRSC—as being novel, despite having been convincingly shown in previous publications (Powell et al, 2020; Franco & Goard, 2021).

2. The authors argue that posterior RSC shows “a clear retinotopic organization” (line 202), but I could not see a clear retinotopic organization in Figure S3. Most of the mice only showed visual responses to a single retinotopic location, and certainly nothing like the retinotopic maps seen in Powell et al., 2020. The authors should clarify and explain any discrepancies with prior work.

3. There are a few topics I wish the authors would have covered in the Discussion. First, are the differences between anterior and posterior RSC just due to connectivity, or are there additional cytoarchitectural differences? Should we be considering dRSC to be a single structure or two separate structures? Second, the authors consider allocentric spatial tuning, but is there any reason to think that egocentric tuning (e.g., Alexander et al., 2020; van Wijngaarden et al., 2020) might show differences

across A-P RSC?

Minor:

1. Throughout the paper, the author's often talk about anterior-posterior differences in RSC, but their data really only covers dysgranular RSC, it is not clear that the A-P gradient of responses applies to granular RSC. The authors should make this clear.
2. For waterfall plots in Figure 2B, please plot cross-validated data (i.e., use half the data to find the preferred spatial location for each cell, and the other half for plotting the response). Also, make sure to describe any sorting in the figure legend.
3. I found the plots of Figure 6E and Figure 7A-D hard to read. I would encourage the authors to make the colors more different (e.g., green and magenta) so the difference/overlap is more clear.

Reviewer #3

(Remarks to the Author)

The study by on the retrosplenial cortex (RSC) in mice, focusing on its role in visuospatial processing and navigation. The study combines 2-photon calcium imaging and brain-wide retrograde circuit mapping to investigate the functional and anatomical organization of the RSC. The research reveals that the RSC has distinct anterior-posterior gradients in terms of neural activity and connectivity, which support specialized processing of visual and spatial information. The findings indicate that the RSC is organized in a modular fashion, with different subregions contributing complementary information during navigation. The study provides insights into the circuitry underlying multimodal integration in the RSC, which is crucial for spatial orientation and cognitive processes.

I like the brain-wide mapping of projectivity to dissociate the functional independence/modularity of anterior and posterior RSC. However, the functional characterization of different parts of RSC has several weaknesses. If the authors were able to consolidate the functional aspects of the study, I think this would be a very important paper, and a good candidate for the journal.

1) The biggest issue lies in the behavioral paradigm. While the study integrates two distinct behavioral paradigms to simultaneously assess visual and spatial functions, it lacks sufficient exploration of the potential interactions between these paradigms when probing the respective functional representation of anterior and posterior RSC.

Spatial cognition depends on multiple sensor modalities. However, when assessing the spatial coding properties, the task only involved tactile landmarks on the belt. Is it likely that the weaker spatial properties of posterior RSC neurons would not hold if visual landmarks were also present? If so, then the conclusion would be different from what has been portrayed in the current manuscript.

2) In addition, the authors should characterize the running speed of the animals when presenting the visual stimulus. If the animals were able to move, there would be a discrepancy between the visual update and self-movement. It is possible that there are differential sensitivity to this discrepancy between the anterior and posterior parts of RSC.

3) The authors should comment on how the current results can be reconciled with earlier studies (e.g., Trask et al. cited in the current paper). How can the different visual-spatial functions and projectivity patterns explain the different roles of anterior and posterior RSC in memory?

4) The authors should acknowledge that Powell et al., 2020 (Figure 1) also mentioned the posterior RSC was more visual.

Version 1:

Reviewer comments:

Reviewer #1

(Remarks to the Author)

Thank you for addressing my comments and questions in this revised version. I am happy with the changes made to text and figures and additional data.

I only have one minor comment – please check that all figure panels are clearly described in the figure legend. For instance, there is no reference to the vertical plots in Figure 8A.

Reviewer #2

(Remarks to the Author)

The authors have made a number of improvements to their manuscript and have addressed all of my concerns. I support publication.

Reviewer #3

(Remarks to the Author)

I thank the authors for the additional experiments and analyses performed, which have significantly improved the manuscript. I have no further comments.

Summary of Major Revisions

In response to reviewer concerns, we performed substantial new experimental work and strengthened multiple analyses:

1. New functional experiments

To determine whether tactile cues drive the anterior–posterior spatial gradient, we added:

- A non-tactile treadmill condition, showing that the gradient persists without textured cues.
- A visually immersive VR condition, demonstrating position tuning across modalities and revealing enhanced posterior position coding under visual-dominant conditions.

These results are presented in Supplementary Figures 2 and 4.

2. Validation of thalamic inputs using independent tracers

Because retro-AAV under-labeled the laterodorsal (LD) nucleus, we performed new CTB retrograde tracing and additional AAV injections into postsubiculum, confirming that LD input is strong but poorly labeled by retro-AAV. These data are now included in Supplementary Figure 9, and LD-related quantification has been revised accordingly.

3. Clarification of laminar and subregional sampling

We quantified imaging depth and coverage across dorsal and ventral RSC, demonstrating predominant sampling within layer 2/3 of dorsal RSC. Figure 1C has been updated with an improved anatomical schematic, and the text now clearly limits functional claims to dorsal RSC.

4. Refined analyses of parahippocampal input

We added quantification of dorsal versus ventral subiculum projections to anterior vs posterior RSC, clarified how entorhinal inputs were normalized, and substantially revised our interpretation to emphasize modality-dependent integration rather than hippocampal dominance.

5. Updated retinotopy and figure improvements

We replaced earlier retinotopic plots with phase maps (Supplementary Figure 5), updated several figures for clarity (Figures 6–8), and applied animal-level statistical tests across all functional comparisons.

6. Expanded Discussion

We incorporated reviewer-requested considerations regarding cytoarchitectural gradients, egocentric coding, and methodological differences relative to prior literature.

Reviewer #1 (Remarks to the Author):

This manuscript investigates multimodal information encoding and long-range input connectivity along the rostral-caudal axis of retrosplenial cortex (RSC), revealing gradients in spatial and visual processing and connectivity. The topic is timely, as systematic subregion analyses in RSC are rare, and the data could substantially advance our understanding of RSC's role in sensory, spatial, and mnemonic functions. However, several issues must be addressed to support the authors' conclusions for publication.

1. Rostro-caudal spatial gradient vs. sensory drive: The observed position-related activity may reflect multimodal integration of somatosensory, visual, and self-motion cues, rather than pure "spatial" coding. Considering the use of tactile cues as landmark

on the treadmill, the stronger rostral position-related signals may arise from stronger somatosensory inputs to the rostral, as compared to caudal, RSC rather than a generalised higher spatial content. Indeed, the viral tracing data in Figure 6H suggest stronger somatosensory cortical inputs to rostral RSC. Indeed, the viral tracing data in Figure 6H suggest stronger somatosensory cortical inputs to rostral RSC.

We agree that the position-related activity observed in RSC likely reflects the integration of multiple sensory modalities, including somatosensory, visual, and self-motion cues, rather than purely abstract spatial coding. We now clarify in the revised text that the stronger position tuning observed in anterior RSC may be driven in part by denser somatosensory and motor inputs, as supported by our retrograde tracing data and previous study (Lande et al., 2023). We have revised the Discussion to better reflect this interpretation (**Line 269–287**).

To conclude that the observed difference reflects spatial coding per se, the study could test position-related signals with non-tactile landmarks (visual cues and running alone) to show that the gradient persists, or temper the conclusion to acknowledge that stronger rostral coding could arise from greater somatosensory and motor inputs, as supported by the tracing data.

To directly address this point, we now include new data from two complementary paradigms designed to isolate the contribution of somatosensory input.

First, we analyzed recordings from sessions in which animals ran on a blank treadmill without textured cues. In this tactile-deprived condition, the overall fraction of position-tuned neurons in anterior RSC was reduced, but the anterior–posterior gradient in position coding remained. This suggests that somatosensory input enhances, but does not fully account for, the stronger spatial coding observed in anterior RSC. These data are now presented in **Supplementary Figure 2**.

Second, we recorded RSC activity in a visually immersive virtual reality (VR) environment. Under these conditions, posterior RSC exhibited robust position tuning, while anterior RSC remained strongly responsive. This indicates that the anterior–posterior differences in position coding are not exclusively dependent on somatosensory input and can be shaped by the dominant sensory modality. These findings are now presented in **Supplementary Figure S4** and discussed in a new section of the Discussion (**Line 269–287**).

Together, these results support the view that anterior and posterior RSC subregions are differentially engaged depending on the available sensory cues, and that the anterior–posterior gradient in position coding reflects flexible, modality-dependent integration rather than a simple hierarchical encoding of “spatialness.” We have

tempered our conclusions accordingly and clarified this interpretation throughout the manuscript.

2. Underestimated thalamic input: The use of retrograde viral tracing and whole-brain quantification of input connectivity provides a rich and informative dataset that complements the functional data. However, there is a major concern regarding thalamic connections: the results show minimal laterodorsal (LD) nucleus input, which contradicts extensive literature identifying the LD as one of the key thalamic afferents to the RSC.

Since this discrepancy may stem from viral tropism, I suggest validating the results with an alternative viral or structural retrograde tracing method before drawing firm conclusions about connectivity.

We thank the reviewer for raising this important point. Upon re-examining our dataset, we confirmed that retrograde AAV injections into both anterior and posterior RSC resulted in only sparse labeling of cell bodies within the laterodorsal (LD) thalamic nucleus. Instead, the observed fluorescence signal in LD primarily reflected passing axonal fibers from RSC projection neurons, rather than retrogradely labeled somata (**Figure 8B**; see also representative images from all animals in **Reviewer Figure 1 below**).

To evaluate whether this underrepresentation reflects a limitation of the viral tracer, we conducted additional control experiments. First, injections of the non-viral retrograde tracer cholera toxin subunit B (CTB) into the RSC resulted in robust LD labeling, consistent with previous anatomical studies and confirming strong LD → RSC connectivity. Second, we injected the same retro-AAV construct (pAAV-CAG-GFP) into the postsubiculum, a region also known to receive dense LD input. These injections similarly failed to produce substantial LD labeling, further suggesting that the lack of labeling reflects viral tropism or transport inefficiency rather than the absence of anatomical connectivity. We have added these new control data as **Supplementary Figure 9**, removed LD, IAM, and IAD data from the main figure and quantification (Figure 8B), and explicitly acknowledged this limitation in the Results section (**Line 239–249**).

Reviewer Figure 1. Retrograde AAV projection patterns from ATN to RSC

Example confocal images show two coronal slices per animal. Slice 1 (more anterior) and Slice 2 (more posterior) illustrate the sparse labeling in LD (yellow areas)

3. Sampling across RSC layers and subregions: Since a major finding of the study is rostro-caudal gradient in spatial and visual information processing, it is important to demonstrate similar sampling across the subregions – granular (gRSC) vs dysgranular (dRSC) areas – and cortical layers (2/3 vs 5) along the rostro-caudal axis. For instance, rostro-caudal differences in cortical layer depth and dRSC/gRSC boundaries may lead to oversampling from gRSC in more rostral regions or oversampling of layer 5 in more caudal regions (300 μ m depth is well within layer 5 in caudal RSC). This issue applies to both functional and anatomical data, and can be addressed by showing the laminar

and regional distribution of imaged neurons and viral injection spread for both rostral and caudal sections to demonstrate similar sampling.

We thank the reviewer for raising this important point regarding potential sampling biases across RSC layers and subregions. We have taken several steps to address this concern in both our anatomical and functional datasets.

For the anatomical tracing experiments, we now quantify the normalized fluorescence density surrounding each injection site to assess the spatial extent of viral expression (**Figure 6D**). In both anterior and posterior injection sites, viral spread encompassed both dorsal and ventral RSC (RSCd and RSCv). However, we note that in one animal (Animal #2), the posterior injection site was slightly anteriorly displaced, resulting in lower labeling from visual cortical areas.

To assess laminar and subregional sampling in our two-photon imaging experiments, we estimated the anatomical locations of imaged neurons by aligning optical planes across sessions to a standardized depth reference. We approximated the first imaging plane to be $\sim 50 \mu\text{m}$ below the pial surface (corresponding to $\sim 300 \mu\text{m}$ below Bregma), with an inter-plane spacing of $\sim 80 \mu\text{m}$. Based on these estimates, the majority of recorded neurons in RSCd were located in layer 2/3—approximately 80% in anterior fields of view and 70% in posterior—while the remaining neurons were distributed in deeper layers. Overall, we estimate that $\sim 80\text{--}90\%$ of functionally imaged neurons were in RSCd, with $\sim 10\text{--}20\%$ in RSCv and no sampling of RSCagl. However, given the limitations of indirect anatomical registration, exact laminar and subregional assignments remain approximate.

We now address these sampling considerations for both the functional and anatomical datasets in the revised Discussion (**Lines 304–318**).

4. *Subicular and ENT input, position coding, and Figure 7G-H: The discussion section suggests that the position-related signal in RSC is hippocampal in origin. If so, one would predict that rostral RSC's stronger position coding should coincide with stronger input from subiculum (the hippocampus's primary output). Instead, Figure 8 shows a posterior bias in subicular input for 3/4 animals (positive bias index). Similarly, although the text states that entorhinal cortex (ENT) input is stronger to anterior RSC, the mean ENT bias index is near zero, and individual animals have both positive and negative biases. It is unclear how the indices in Figure 7H align with the stacked bar chart in Figure 7G, since the panel captions lack detail.*

The manuscript also describes a dorsal–ventral gradient in subicular projections to anterior vs. posterior RSC, yet no per-animal or group-level quantification is provided. If Figure 7G illustrates this gradient, please explain the panel in the caption to define all colours and results clearly (it's very difficult to understand this panel as is), and include statistics (mean \pm SEM, test values, p-values) to support any regional differences.

Overall, the claim that rostro-caudal differences in position coding are explained by parahippocampal connectivity is not fully supported, because i) the gradient may reflect sensorimotor integration rather than hippocampal-dependent spatial coding (see comment 1); ii) it is unclear whether subiculum itself exhibits a comparable dorsal–ventral gradient in position coding (please cite any relevant studies); iii) a significant difference in dorsal vs. ventral subicular input to RSC has not been convincingly demonstrated.

Role of parahippocampal input in position coding

We agree that anterior–posterior differences in position coding cannot be attributed solely to parahippocampal connectivity. In the revised Discussion (**Line 269–287**), we emphasize that multimodal integration may play a role in shaping position-tuned responses in RSC, particularly in the anterior subregion. This view is supported by recent studies (e.g., Gianatti et al., 2023; Lande et al., 2023), which demonstrated that brain areas such as M2 and PPC convey robust position-related signals, especially in tactile-rich environments.

Regarding hippocampal connectivity, prior work has established a dorsal–ventral functional gradient within the subiculum: dorsal subiculum (the main output of dorsal CA1) is implicated in spatial and cognitive processing, while ventral subiculum (output of ventral CA1) is more involved in emotional and stress-related functions (Fanselow & Dong, 2010). Spatially selective cells—such as boundary vector cells (Lever et al., 2009) and geometry-sensitive neurons (Sun et al., 2024), are preferentially located in dorsal subiculum. In our retrograde tracing experiments, we observed that while the overall fraction of subicular inputs was slightly higher for posterior RSC, clear topographic patterns were observed: dorsal subiculum preferentially targets anterior RSC, whereas ventral subiculum sends stronger projections to posterior RSC. To

support this interpretation, we now provide quantification, statistics, and representative confocal images in the **Figure 8A**.

Clarification of ENT input and figure interpretation

We appreciate the reviewer's comment regarding the interpretation of entorhinal cortex (ENT) input. Although the average bias index across animals is close to zero, we find that when ENT inputs are normalized within the hippocampal/parahippocampal group, a greater proportion is directed to anterior RSC. To clarify this distinction and avoid confusion, we have revised the results, figure legend and improved the visual presentation to better illustrate the data.

- **Horizontal bar plots** show the fraction of inputs from each subregion, normalized by the total number of labeled neurons across all subregions within that region (e.g., across all parahippocampal subregions). This provides a measure of the percentage contribution of each subregion to RSC inputs (e.g., ~50% of inputs from the parahippocampal region to anterior RSC arise from ENT)
- **Vertical bar plots and bias indices** are calculated relative to the total input population, capturing the absolute proportion of each input source (e.g., ENT accounts for ~3% of the total inputs to anterior RSC).

Together, these complementary analyses convey both subregional specificity and overall distribution biases.

Regarding statistics, due to the limited sample size ($n = 4$), we report mean \pm SEM and show paired data across animals, but do not perform formal statistical comparisons.

Additional comments:

• Powell et al. (2020) demonstrated a rostro-caudal gradient in visual responses in RSC. Please discuss and contrast their findings in the discussion section on visual properties. The text currently implies that no prior study has explored this question in the context of spatial and visual processing and that prior knowledge about rostro-caudal organization was limited to object recognition and contextual fear (lines 421–427).

We revised Discussion section (Line 288–302) to contrast our observations with those Powell et al. (2020).

Page 3, line 91: The description of imaged areas is unclear and does not match Figure 1C. The text describes imaging "in RSPd within the agranular layer of the RSC at depths of 60–360 μ m below the pial surface." Dysgranular and agranular RSC are

two separate but adjacent subregions. Additionally, Figure 1C shows the FOV covering both dysgranular (RSCd) and part of granular RSC (referred to as RSCv in this manuscript). Finally, based on the Allen Brain Atlas, this imaging depth covers both layer 2/3 and layer 5 (especially in caudal RSC, as explained above). Please clarify the imaging location with details about layers and subregions in both rostral and caudal RSC and resolve the inconsistency between the text and Figure 1C.

As noted in our response to point 3, we have revised Figure 1C by removing “RSCv” label and adding a sagittal schematic to illustrate the estimated imaging depth, providing clearer context for laminar coverage. These changes are now reflected in the main text and updated figure legend

For all statistical analyses, neurons across all animals are pooled and treated as independent replicates. This inflates N and raises the risk of type I errors (small effects may appear significant even if between-animal variance is large). I recommend either collapsing each animal’s anterior and posterior responses into per-animal summary statistics and testing across animals, or fitting a mixed-effects model with animal as a random effect.

Most statistical comparisons of neuronal activity across conditions (e.g., Figures 2D–2G) were already performed at the animal level, using per-animal means as independent observations (**Line 746–752**). Visual response reliability (Figure 4B) and direction selectivity index (Figure 5F) now follow the same procedure, ensuring consistency across all metrics. These revisions ensure that all reported statistical tests indicating significance are now based on animal-wise comparisons, rather than treating individual neurons as independent replicates. This approach mitigates the risk of inflated Type I errors and more appropriately accounts for between-animal variability.

Two different transgenic mouse lines are used in this study. What was the rationale, and are the results consistent between these two lines?

The initial experiments were performed in the Thy1-GCaMP6s line which was used in our observation of position-related activity in mouse RSC (Mao et al 2017). We also recorded from CaMKII-tTA × TRE-GCaMP6s which provides distinct sampling and broader coverage of excitatory neurons. Despite differences in expression density, both lines exhibited the same anterior–posterior trends in position-related and visually evoked responses. Because our key comparisons were performed within animals, potential between-line variability was minimized. To increase transparency, we revised the figures with animal-wise quantification to display the two lines using distinct color codes (Thy1: gray; CaMKII: black).

Page 12, lines 306-309: Please include SEM values.

We have added the corresponding SEM values (fraction of cortical RSC-projecting neurons)

Page 13, lines 318 -322, lines 330- 335, lines 339 – 342: Statistics are missing in all these sections that conclude differences between anterior and posterior RSC connectivity

Given the limited sample size ($n = 4$ animals), we decided not to perform formal statistical tests for these connectivity comparisons, as they would be underpowered and potentially misleading. Instead, we present mean \pm SEM values and connect paired data points across animals in the figures to allow readers to visually assess consistency across animals.

Page 13, line 328: revise “anterior dorsal” to “anterior” thalamic nuclei (ATN)

We have revised the terminology in the main text to “anterior thalamic nuclei (ATN)” to align with common convention. In the figure captions for Figures 6–8, we retained the term “anterior group of the dorsal thalamus” to remain consistent with the nomenclature used in the Allen Mouse Brain Atlas (Wang et al., 2020).

Reviewer #2 (Remarks to the Author):

In this manuscript, Wei et al. investigate differences in connectivity and functional responses between anterior and posterior dysgranular retrosplenial cortex (dRSC). They do some convincing two-photon imaging experiments to show that anterior and posterior dRSC show differences in visual and allocentric spatial tuning. They also do some nice histology using retrograde viral tracing to quantify differences in connectivity. These findings, though perhaps of somewhat specialized interest, would make a nice contribution to the field. However, I do have a few concerns about specific analyses and their scholarship, as described below:

Major:

1. The scholarship is not very thorough when discussing differences in anterior and posterior connectivity. For example, in the introduction they cite papers that have nothing to do with anterior-posterior gradients of visual/hippocampal input (e.g., Niell and Stryker, 2008, Goldbach et al., 2021, etc.), while skimming over or missing papers of very direct relevance. For example, Yamawaki et al., J Neuro, 2016 and Powell et al, Cereb Cortex, 2020 (both cited elsewhere in the manuscript, but not here) both offer direct evidence of anatomical and functional gradients along anterior-posterior RSC. The authors also completely missed Franco & Goard, Science Advances, 2021, which show both anatomical and functional gradients in anterior-posterior RSC using similar techniques.

As a result, the authors present certain results—such as stronger visual tuning in posterior dRSC—as being novel, despite having been convincingly shown in previous publications (Powell et al, 2020; Franco & Goard, 2021).

We thank the reviewer for this valuable feedback. We have revised the Introduction to more accurately situate our study within prior work on anterior–posterior organization of RSC (**Line 47–57**). In addition, we have tempered our novelty claims: the manuscript now explicitly acknowledges that stronger visual responsiveness in posterior RSC has been reported previously (Powell, 2020; Franco & Goard, 2021). Our contribution lies in extending these findings by (i) jointly comparing spatial and visual coding within the same animals across anterior vs posterior RSC, (ii) identifying a novel spatiotemporal tuning gradient (posterior: fine/slow motion; anterior: coarse/fast motion), (iii) testing modality dependence with non-tactile, dark, and VR landmark–rich conditions, and (iv) linking these functions to brain-wide input mapping.

2. The authors argue that posterior RSC shows “a clear retinotopic organization” (line 202), but I could not see a clear retinotopic organization in Figure S3. Most of the mice only showed visual responses to a single retinotopic location, and certainly nothing

like the retinotopic maps seen in Powell et al., 2020. The authors should clarify and explain any discrepancies with prior work.

To enhance the visualization of retinotopic organization in the RSC, we have updated **Supplementary Figure 5** to display phase maps, which depict each pixel's preferred stimulus location independent of response amplitude. Additionally, we now include a **supplementary video** to illustrate how retinotopic signals extend from lateral visual areas into the RSC. Across most animals, we observed a consistent spatial pattern in which retinotopic responses followed a posterolateral-to-anteromedial axis, corresponding to the upper-to-lower nasal visual field. In two animals (#3 and #7), this pattern was less distinct, which may reflect their reduced locomotor activity during imaging. This is consistent with findings from Powell et al. (2020), who reported that visual responses in RSC neurons are enhanced during periods of increased locomotion.

Powell et al. also described azimuthal gradients of cellular responses in the posterior RSC, which were less apparent in our widefield imaging data. This discrepancy may stem from methodological differences: Powell et al. employed drifting bar gratings optimized for continuous retinotopic mapping, whereas we used discrete circular patch stimuli, which are less sensitive to fine azimuthal gradients. Additionally, widefield imaging captures population-level activity, potentially masking the finer retinotopic structure revealed by cellular-resolution approaches (**Line 288–302**).

3. There are a few topics I wish the authors would have covered in the Discussion. First, are the differences between anterior and posterior RSC just due to connectivity, or are there additional cytoarchitectural differences? Should we be considering dRSC to be a single structure or two separate structures? Second, the authors consider allocentric spatial tuning, but is there any reason to think that egocentric tuning (e.g., Alexander et al., 2020; van Wijngaarden et al., 2020) might show differences across A-P RSC?

We thank the reviewer for these insightful points. We now address both in the revised Discussion: cytoarchitectural considerations and functional gradients in RSC (Lines 304–318), and the potential for egocentric coding differences along the anterior–posterior axis (Lines 320–329).

Minor:

1. Throughout the paper, the author's often talk about anterior-posterior differences in RSC, but their data really only covers dysgranular RSC, it is not clear that the A-P gradient of responses applies to granular RSC. The authors should make this clear.

We agree with the reviewer and have clarified this point in the revised Discussion (Lines 304–318). To illustrate the subregional and laminar sampling more clearly, we

also added an estimated depth distribution of imaged neurons in **Figure 1C**, highlighting that our functional data primarily reflect dorsal RSC (RSCd)

2. For waterfall plots in Figure 2B, please plot cross-validated data (i.e., use half the data to find the preferred spatial location for each cell, and the other half for plotting the response). Also, make sure to describe any sorting in the figure legend.

We thank the reviewer for the helpful suggestion. For the population trial-averaged activity plots (**Figure 2B, Supplementary Figure 2A–B, Supplementary Figure 4D**), we now apply a cross-validation procedure: one half of the trials is used to determine each neuron's preferred spatial location for sorting, and the other half is used to plot the heatmap. The persistence of the diagonal pattern in these cross-validated plots confirms that the observed spatial tuning is robust and not an artifact of sorting. The figure legends have been updated accordingly to clarify this analysis.

3. I found the plots of Figure 6E and Figure 7A-D hard to read. I would encourage the authors to make the colors more different (e.g., green and magenta) so the difference/overlap is more clear.

We have improved the visualization in **Figures 6 and 7 (now are Figures 6–8)** by simplifying the color coding to better match the physiological data and by enhancing the contrast between groups. In Figure 7, we further increased clarity by splitting the reconstructions into anterior versus posterior projections and by replacing the stacked bar chart with horizontal bar plots. These changes make the overlap and regional differences easier to interpret

Reviewer #3 (Remarks to the Author):

The study by on the retrosplenial cortex (RSC) in mice, focusing on its role in visuospatial processing and navigation. The study combines 2-photon calcium imaging and brain-wide retrograde circuit mapping to investigate the functional and anatomical organization of the RSC. The research reveals that the RSC has distinct anterior-posterior gradients in terms of neural activity and connectivity, which support specialized processing of visual and spatial information. The findings indicate that the RSC is organized in a modular fashion, with different subregions contributing complementary information during navigation. The study provides insights into the circuitry underlying multimodal integration in the RSC, which is crucial for spatial orientation and cognitive processes.

I like the brain-wide mapping of projectivity to dissociate the functional independence/modularity of anterior and posterior RSC. However, the functional characterization of different parts of RSC has several weaknesses. If the authors were able to consolidate the functional aspects of the study, I think this would be a very important paper, and a good candidate for the journal.

1. The biggest issue lies in the behavioral paradigm. While the study integrates two distinct behavioral paradigms to simultaneously assess visual and spatial functions, it lacks sufficient exploration of the potential interactions between these paradigms when probing the respective functional representation of anterior and posterior RSC.

Spatial cognition depends on multiple sensor modalities. However, when assessing the spatial coding properties, the task only involved tactile landmarks on the belt. Is it likely that the weaker spatial properties of posterior RSC neurons would not hold if visual landmarks were also present? If so, then the conclusion would be different from what has been portrayed in the current manuscript.

We thank the reviewer for this important point. To directly address this concern, we performed an additional set of experiments in which mice navigated a visually immersive virtual reality (VR) environment. In this task, salient visual landmarks were the dominant spatial cues, in contrast to the tactile cued belt paradigm used previously. These new data, now included in **Supplementary Figure 4**, show that posterior RSC exhibits robust position-related activity in the presence of visual landmarks, with a clear sequence of spatial tuning across neurons. Notably, anterior RSC continued to show strong spatial coding in the VR task, consistent with its broader integrative role. However, compared to the tactile condition, posterior RSC position coding was significantly enhanced under visually enriched conditions, supporting the idea that its spatial engagement is more dependent on visual context.

We now explicitly incorporate these results into the revised manuscript. In the Discussion, we highlight this distinction, emphasizing that anterior RSC encodes

spatial information across both tactile and visual conditions, while posterior RSC is more selectively recruited when structured visual input is available. This suggests that position coding in RSC is shaped by the dominant sensory modality, and that anterior and posterior subregions contribute differentially depending on sensory context.

These findings strengthen the central message of the manuscript: that the anterior-posterior axis of RSC exhibits distinct patterns of multimodal integration, rather than uniform spatial representation across tasks. We have revised the text throughout the Results and Discussion to clarify this point and avoid overgeneralization from the tactile paradigm alone.

2. In addition, the authors should characterize the running speed of the animals when presenting the visual stimulus. If the animals were able to move, there would be a discrepancy between the visual update and self-movement. It is possible that there are differential sensitivity to this discrepancy between the anterior and posterior parts of RSC.

To address this point, we have included a new supplementary figure (**Supplementary Figure 8**) quantifying both the animals' running behavior and RSC population neural activity during visual stimulation. Animals exhibited a slight reduction in running speed during visual stimulus presentation, particularly in response to temporal-to-nasal motion. This suggests that visual motion can influence ongoing locomotor behavior, potentially reflecting a mismatch between visual flow and self-motion cues.

To assess whether such mismatch impacts neural activity, we analyzed the amplitude of population calcium signals ($\Delta F/F_0$) across sessions for opposing visual motion directions. As shown in Supplementary Figure 7C, we did not observe consistent differences in response amplitude across matched stimulus orientations, indicating that visual-motor mismatch does not strongly modulate overall RSC population activity.

To further determine whether self-motion alone might account for responses attributed to visual stimulation, we computed a speed score for each neuron—defined as the correlation between instantaneous deconvolved activity and the animal's running speed (Kropff et al., 2005; Supplementary figure 7D). We identified both positively and negatively speed-modulated neurons in both anterior and posterior RSC. The distribution of speed scores was slightly broader in anterior RSC, consistent with our anatomical tracing results showing stronger motor-related inputs to this region.

We next assessed the overlap between speed-modulated and visually responsive neurons. As shown in Supplementary Figures 7E–F, the two populations were largely non-overlapping, indicating that visual and locomotor signals are encoded by distinct neuronal subpopulations within RSC.

Together, these results suggest that although visual stimuli can influence running behavior, this does not lead to systematic changes in population response amplitude. Furthermore, the minimal overlap between speed- and visually modulated neurons supports the conclusion that visually evoked activity in RSC is not confounded by self-motion signals at the single-cell level.

3. The authors should comment on how the current results can be reconciled with earlier studies (e.g., Trask et al. cited in the current paper). How can the different visual-spatial functions and projectivity patterns explain the different roles of anterior and posterior RSC in memory?

We thank the reviewer for highlighting this connection. While our study focuses on visuospatial processing rather than memory per se, we agree that our findings provide a useful framework for interpreting prior work on RSC's role in memory. For instance, Trask et al. found that posterior RSC is required for encoding and retrieval of contextual memories, whereas anterior RSC is more involved when integrating context with event-related information. This aligns with our observation that posterior RSC is more visually driven and context-sensitive, while anterior RSC is more integrative and position-reliable. We have added a brief note on this in our response to reviewers but did not expand it in the main text due to the differing behavioral paradigms.

4. The authors should acknowledge that Powell et al., 2020 (Figure 1) also mentioned the posterior RSC was more visual.

In revised Discussion, we explicitly state that our observation of stronger visual responses in posterior RSC is consistent with Powell et al. (2020), who also reported enhanced visual responsiveness in posterior RSC.

Reviewer #1:

Thank you for addressing my comments and questions in this revised version. I am happy with the changes made to text and figures and additional data.

Response: We thank the reviewer for their positive evaluation of our revisions.

I only have one minor comment – please check that all figure panels are clearly described in the figure legend. For instance, there is no reference to the vertical plots in Figure 8A.

Response: We thank the reviewer for catching this omission. We have carefully reviewed all figure legends to ensure that every panel is clearly described. We have revised the Figure 8A legend to include a description of the vertical bar plots showing density and fraction of inputs.

Reviewer #2:

The authors have made a number of improvements to their manuscript and have addressed all of my concerns. I support publication.

Response: We thank the reviewer for their support and positive assessment of our manuscript.

Reviewer #3:

I thank the authors for the additional experiments and analyses performed, which have significantly improved the manuscript. I have no further comments.

Response: We thank the reviewer for their thorough evaluation and for acknowledging the improvements made to the manuscript.